# The seasonal evolution of albedo across glaciers and the surrounding landscape of the Taylor Valley, Antarctica

Anna Bergstrom[1], Michael Gooseff [2], Madeline Myers[3], Peter T. Doran[3], and Julian Cross[4]

[1]Department of Geological Sciences, University of Colorado-Boulder, 80305, United States
[2]Department of Civil Environmental and Architectural Engineering, University of Colorado-Boulder, 80305, United States
[3]Department of Geology and Geophysics, Louisiana State University, Baton Rouge, 70803, United States
[4]Department of Geology, Portland State University, Portland, 97201, United States

*Correspondence to*: Anna Bergstrom (anna.bergstrom@colorado.edu)

**Abstract.** The McMurdo Dry Valleys (MDVs) of Antarctica are a polar desert ecosystem consisting of alpine glaciers, ice-covered lakes, streams, and expanses of vegetation-free rocky soil. Because average summer temperatures are close to 0°C, glacier melt dynamics in particular, but the MDV ecosystem in general, are closely linked to the energy balance. A slight increase in incoming radiation or change in albedo can have large effects on the timing and volume of melt water. However, the seasonal evolution or spatial variability of albedo in the valleys has yet to fully characterized. In this study, we aim to understand the drivers of landscape albedo change within and across seasons. To do so, a box with a camera, GPS, and short-wave radiometer was hung from a helicopter that flew transects 4-5 times a season along Taylor Valley. Measurements were repeated over three seasons. These data were coupled with incoming radiation measured at 6 meteorological stations distributed along the valley to calculate the distribution of albedo across individual glaciers, lakes, and the soil surfaces. We hypothesized that albedo would decrease throughout the austral summer with ablation of snow patches and increasing sediment exposure on the glacier and lake surfaces. However, small snow events (< 6 mm water equivalent) coupled with ice whitening caused spatial and temporal variability of albedo across the entire landscape. Glaciers frequently followed a pattern of increasing albedo with increasing elevation as well as increasing albedo moving from east to west laterally across the ablation zone. We suggest that spatial patterns of albedo are a function of landscape morphology trapping snow and sediment, longitudinal gradients in snowfall magnitude, and wind-driven snow redistribution from east to west along valley. We also compare our albedo measurements to the MODIS albedo product and found that overall the data have reasonable agreement. The mismatch in spatial scale between these two datasets results in variability, which is reduced after a snow event due to albedo following valley-scale gradients of snowfall magnitude. These findings highlight the importance of understanding the spatial and temporal variability in albedo and the close coupling of climate and landscape response. This new understanding of landscape albedo can constrain landscape energy budgets, better predict melt water generation on from MDV glaciers, and how these ecosystems will respond to changing climate at the landscape scale.

# 1 Introduction

In most regions of the cryosphere, the absorption of short wave radiation is the main source of energy for snow and ice ablation (Male and Granger, 1981). Therefore albedo, the proportion of incoming radiation that is reflected is an important parameter to accurately measure. Albedo for fresh snow can be very high, but decreases as snow melts, ice grains age and metamorphose, and dust, debris, or water is accumulated on the surface (Grenfell, 2004). Over the course of a season, there can be very rapid and spatially variable changes in albedo, which feeds back on the degree of melt water generation (e.g. Perovich & Polashenski, 2012).

During the austral summer in McMurdo Dry Valleys (MDVs) of Antarctica, the constant solar radiation and near freezing average temperatures mean that any small change in solar radiation or albedo can cause large changes in glacial melt (Fountain et al., 1999). Glacial melt is the main source of water to ephemeral streams, the soil zone near the streams, and closed basin lakes. For this reason, it is important to correctly predict the degree of melt in order to understand ecosystem response and function. Beyond meltwater generation, the ecological processes across the entire landscape are regulated by incoming solar radiation, spatial distribution of snow and liquid water, and soil and sediment temperatures. For example, the light reflected by the permanently ice-covered lakes, has a large effect on the primary productivity of organisms in the lake water column below (Fritsen and Priscu, 1999). Over 24 years of monitoring, Gooseff et al. (2017) observed changes to both physical and biological properties across the MDV ecosystem, especially in response to the 2001-02 'flood year' in which the highest glacial melt on record occurred.

Extensive modeling of ablation on the MDV glaciers has suggested that there may be long-term changes in albedo (Hoffman et al., 2014, 2016). Hoffman et al. (2016) developed a spatially distributed glacier energy balance model, which consistently under-predicted ablation in the years after 2006. They found that a -0.09 adjustment in the albedo parameter reduced model bias as much as a +6°C adjustment in temperature. This adjustment in albedo is within the realm of seasonal and spatial variability, while a 6°C change in temperature is beyond predicted warming over the next 100 years over the Antarctic continent (Chapman and Walsh, 2007). While the exact mechanism and timing of this shift is unclear, Hoffman et al. (2016) hypothesize that a result was a change in the physical properties of the glacier, specifically albedo, which they suggest was driven by increased sediment on glacier surfaces. This shift is hard to detect because the only observations of albedo, also used for model parameterization, are made at meteorological stations. These stations are generally located at higher elevations in the ablation zones on the 'cleaner' parts of the glaciers. It is unknown how representative these point albedo measurements are of the entire glacier ablation zone.

Albedo is typically measured directly, via remote sensing, or modeled numerically. Each of these approaches are well accepted but have unique shortcomings. Local albedo measurements are made at a point, making it difficult to generate a high enough density of measurements to create an accurate spatial coverage of a parameter that is highly spatially variable (Grenfell, 2004). Remote sensing is limited by spatial resolution, the number of cloud-free days over which the measurement can be made, and may have errors due to the atmospheric absorption (Stroeve et al., 1997, 2005). Numerical models are typically

based on experimentally derived parameters and may be poorly linked to the true physical properties of the surface being modeled (Gardner and Sharp, 2010).

The glaciers of the MDVs are generally small, only a few kilometers wide, and due to the high latitude (76° S) resulting in increased atmospherically-derived error, satellite-based remote sensing is not a practical option for estimating albedo in this setting and for this application. Current estimates of albedo are made at only a few points in locations that do not represent the full range of surface types in the valley, particularly on the glaciers, and is not measured over lake ice or soil at all. The primary goal of this study is to characterize the spatial and temporal variability of albedo of the meltwater-generating ablation zones of MDV glaciers throughout the summer melt season. To do so, we use high-resolution airborne reflected radiation data coupled with measured incoming radiation at a network of meteorological stations throughout the valley. The approach also collects data not just on the glaciers but across the entire valley, which allows us to interrogate patterns of albedo for the permanently ice-covered lakes and soils and evaluate the spatial patterns of response to snow events. This research will help us better understand the spatial sources of melt on the glaciers and contribute to better parameterization of energy balance models not only on the glaciers but across the MDVs.

## 2 Site Description and Methods

### 2.1 The Taylor Valley

The McMurdo Dry Valleys are the largest ice free area on the Antarctic continent (Levy, 2013). They are bounded by the Transantarctic Mountains to the west and the Ross Sea to the east. The landscape is composed of piedmont and alpine glaciers, permanently ice-covered lakes on the valley floors, and vegetation-free dry rocky soils over permafrost, with localized areas of buried ice.

During the winter (March-September), there is little to no incoming solar radiation and temperatures average -30°C (Fountain et al., 1999). The MDVs also experience foehn wind events, very strong winds draining off the polar plateau moving down valley. During a foehn event, air temperatures increase by as much as 30°C and wind speeds reach up to 20 m s$^{-1}$ (Nylen et al., 2004). Foehn winds can occur any time of year but are most frequent and dramatic in winter and can cause massive sublimation of ice and snow. In summer (December- February) there is constant sunlight and temperatures average -7°C. There is very little precipitation in the valleys. Snow events are generally only a few centimeters, are quickly redistributed by wind, and can rapidly sublimate.

This study focuses on the Taylor Valley. The head of the valley is filled by the Taylor Glacier, which flows from the East Antarctic ice sheet (Figure 1). The valley is bounded by the Kurki Hills to the south and the Asgard Mountain Range to the north. Cold-based alpine glaciers flow out of both ranges toward the valley floor (Chinn, 1998). The largest two glaciers, Canada and Commonwealth originate in the Asgard Mountains and reach the valley floor, terminating in 20 m tall cliffs (Figure 1). Glaciers, particularly those that reach lower elevations, receive wind transported material that melts the ice below it and sinks in to form cryoconite holes (1-100 cm diameter) or larger melt basins (5-20m diameter). There is a distinct visible

difference in glacier morphology on the up-valley side of the Canada and Commonwealth glaciers that are exposed to foehn winds and therefore have increased potential for sediment deposition (Figure 1, Sabacka et al., 2012).

There are three main closed-basin permanently ice-covered lakes: Lake Bonney, Lake Hoare, and Lake Fryxell (Figure 1). The lakes also receive sediment from aeolian deposition; and due to coupled high melt and sublimation rates, have
highly variable roughness and ice morphology. Moats frequently melt out around the lake edges during the austral summer which refreeze to smooth ice in the fall. The permanent ice can develop melt ponds and other ablation features which can be several meters in area and reach over a meter of vertical relief. Over the course of work on the MDVs, researchers have anecdotally observed that lakes follow cycles of sediment accumulation and ablation feature development. Lakes receive water from more than 20 ephemeral proglacial streams and direct melt from glaciers. The rest of the valley is made up of bare, rocky
soil and exposed bedrock.

## 2.2 Instrumentation and Data Collection Flights

This study takes a helicopter-based remote sensing approach to measure landscape albedo. We manufactured a weighted box with a fin to reduce swing during a flight (Figure 2a). Underneath the box, we mounted a shortwave radiometer (CMP3, Kipp and Zonen) and camera (CC5MPX, Campbell Scientific). A gps unit (GPS16X-HVS, manufactured by Garmin
for Campbell Scientific) was fixed to the top of the box. All instruments were wired in to a data logger (CR1000, Campbell Scientific) and programmed to collect data every three seconds. Before each flight, the radiometer and camera lens were cleaned. The box with instrumentation was attached to a helicopter with a 6 m lead line.

Flights originated at Lake Hoare camp. Pilots flew east over Canada Glacier, Lake Fryxell, and Commonwealth Glacier. Pilots were instructed to fly off Commonwealth glacier to the east, turn around, and return over Commonwealth
Glacier, the southern side of Lake Fryxell, and the lower elevations of Canada Glacier. They would then continue west to Lake Bonney and Taylor Glacier, turning around at the meteorological station located on Taylor Glacier and returning to Lake Hoare camp (Figure 1). We instructed pilots to fly 100 m off the ground at a speed of 50 knots and to fly as directly as possible over the meteorological stations located on Canada, Commonwealth and Taylor Glaciers.

Five flights were made over the 2015-16 season (20 Nov 15; 7 Dec 15; 24 Dec 15; 5 Jan 16; 12 Jan 16), five flights
over the 2016-17 season (11 Nov 16; 3 Dec 16; 14 Dec 16; 3 Jan 17; 23 Jan 17), and four flights over the 2017-18 season (21 Nov 17; 6 Dec 17; 27 Dec 17; 13 Jan 18; Bergstrom and Gooseff, 2019). Solar zenith angles ranged from 54.3 to 61.8° across all flights. We attempted to schedule flights on completely cloud-free days or homogenously cloudy days to reduce variability in incoming radiation across the valley (Figure 3). Flight protocols did not allow for scientists to fly with the pilots during these data collections (because of the sling load under the aircraft). All helicopter pilots were given the same explicit directions
for flight lines. Some variability in each flight, due to flight scheduling and variability in incoming radiation or variability in flight paths was unavoidable.

## 2.3 Post-processing

All spatial analysis was performed using the ArcGIS software package (version 10.7). True color images and radiation data were examined visually to help develop a spatially-based classification scheme. All data collected in a flight were classified. Individual measurements were either discarded if they did not meet usability standards or were associated with a given landscape feature and the closest meteorological station. Data were discarded if they were collected at the beginning and end of the flight as the instruments were being picked up or set down by the helicopter. Data were also discarded if they were too close to the edge of a given landscape feature.

At the edge of a feature, the radiometer is receiving reflected radiation from multiple landscape types, and the measurement cannot be attributed to a specific landscape feature. Outlines of all glaciers and lakes were manually delineated using Landsat8 true color imagery (collected 14 January, 2016, Figure 2b, c). We then calculated a 100 m buffer from the edge of each landscape feature. The 100 m distance was determined by cross-comparing a range of buffer distances and the relative change in radiation with distance to the edge. The rate of change of reflected radiation dropped off considerably greater than 100 m from the edge of a glacier or lake, and we assume that change in reflected radiation is representative of true variability of that surface type.

Each point over a given landscape feature was coded to associate it with one of four lakes (Bonney, Mummy Pond, Hoare, or Fryxell), or one of five glaciers (Taylor, LaCroix, Suess, Canada, or Commonwealth) (Figure 2 b, c). Mummy Pond and LaCroix and Seuss glaciers are small and were not consistently measured. Data from these locations are used in a lumped cross-landscape type comparison (i.e. glaciers vs. lakes) but were not used in an individual analysis (i.e. Canada Glacier vs. Taylor Glacier) due to lack of sufficient data. Data collected over soil surfaces were separated in to two halves of the Taylor Valley: the Bonney and Hoare basins (referred to as the west soils) and the Fryxell basin (referred to as the east soils). Flight paths varied, which led to inconsistent numbers of measurements across flights. Glaciers are the main focus of this study and we frequently collected over 80 useable measurements on a given flight for one glacier with a minimum of 48 (Table S1). Data over soil surfaces were collected as the transition from a lake to a glacier rather than explicitly targeted and therefore had the lowest number of measurements across all flights (Table S1). In some cases, pilots flew too close to the edge of a lake, which necessitated discarding measurements that should be classified as a lake and resulted in low total counts of lake measurements (e.g. 7 Dec 15, Table S1).

We used incoming shortwave radiation and snow fall data from meteorological stations located throughout the valley (Figure 1). Lakes Bonney, Fryxell, and Hoare and Canada Glacier meteorological stations measure incoming shortwave radiation with a LI-COR LI-200R pyranometer and Commonwealth and Taylor glacier meteorological station with an Eppley PSP pyranometer. Incoming radiation is sampled every 30 seconds and an average is recorded every 15 min. Stations are leveled and measuring reflected radiation over a level patch of ice. All meteorological stations are visited annually for maintenance and instruments are replaced and recalibrated according to the manufacturers' recommendations. We used snowfall data measured at Lakes Bonney, Hoare, and Fryxell meteorological stations. Precipitation at Lakes Bonney and

Fryxell was measured with a Campbell Scientific SR-50 sonic ranger. Depth is corrected to volume using a snow density of 83 kg m$^{-3}$. The snow density was derived from measurements of freshly fallen snow on a smooth, clean 0.5m x 0.5m surface during December, 2018. A Belfort weighing bucket with a Nipher Shield measures precipitation at Lake Hoare. Daily total accumulation was derived from 15-minute averages. Station set-up and data processing details are explained in depth by Fountain et al. (2010).

We calculated apparent albedo using the incoming radiation measured at the closest meteorological station to a given measurement determined by the Thiessen polygon method (Figure 2b, c). We used the measurement closest to the time the reflected radiation measurement was made. In a few cases, the closest meteorological station was not recording or was under a cloud (Figure 3 a-c) and produced an apparent albedo over 1. In those cases, we calculated an average incoming radiation from all other meteorological stations to calculate albedo. All data analysis and statistics were done in Matlab (Mathworks v. 2018a).

We tested for significant differences between and across landscape types, features, and flights. Because points are collected sequentially along a flight line they are not independent observations and there is inherent spatial autocorrelation. We address this with transect line correlation analysis, a statistical method that uses a randomization technique (Malatesta et al., 1992). Data are first classified as: $n$ points in a category (e.g. Taylor glacier on 3 Dec, 2016) with associated albedo, making data pairs ($X_i$, $Y_i$) where $i = 1,2,...,n$, $X$ is albedo and $Y= 1$ are the measurements belonging to that category and $Y= 0$ are all other measurements (e.g. all points measured on all other glaciers on 3 Dec 2016). We test for the correlation between X and Y, with a null hypothesis that there is no correlation between location/time and albedo. First, we calculate the Pearson correlation coefficient between X and Y. Then albedo is randomly permuted, while the category is held constant (i.e. X is randomly permuted and Y is held constant) and the Pearson correlation coefficient is calculated again. This random permutation is repeated 1,000 times. The percentage of the number of times the correlation coefficient using a random permutation exceeds the correlation coefficient of the original comparison is calculated. This is the confidence level for rejecting the null hypothesis. We use a 95% confidence level to determine if a given category is significantly different (e.g. Taylor glacier has significantly lower albedo than all other glaciers on 3 Dec 2016). This test can be run for any combination of flights and locations such as testing if albedo of Canada Glacier significantly increased from the first to second flight of the 2015-16 season or if all glaciers had significantly higher albedo than lakes and soils on 22 Nov 2017. All results reporting significant differences between features or flights were determined using this method.

**2.4 MODIS data acquisition**

We compare calculated albedos measured by the helicopter to MODIS-derived albedo estimates. The MCD43A3 Version 6 MODIS albedo product provides daily albedo measurements at a 500-m spatial resolution (Schaaf and Wang, 2017).The MODIS instrument, onboard both the Aqua and Terra satellites, images the same location every 1 to 2 days. These satellites have a sun-synchronous, near-polar orbit such that they acquire at least two images of Taylor Valley per day – at roughly 10:30 AM and 1:30 PM local time. The MCD43A3 product deploys a bi-directional reflectance distribution function

(BRDF) to calculate broadband and spectral albedo at local solar noon using a centered, 16-day moving window, and the center day of the window given the highest weight. Albedo is available as 'black-sky' (BSA, direct component) and 'white-sky' (WSA, diffuse component) albedo for seven MODIS narrow and three broad bands.

Daily MCD43A3 shortwave broadband (0.3-5.0μm) images were compiled in Google Earth Engine from each flight day. MCD43A3 albedo values were extracted for the pixel corresponding to the location of each airborne albedo measurement. A 'blue-sky' albedo was calculated using a linear combination of the BSA and WSA the components (Möller et al., 2014). These two components are essentially equal at approximately 50° (Stroeve et al., 2005; <1% on average in this dataset), which corresponds to a typical solar noon zenith angle at in Taylor Valley and is within the range of zenith angles over which the airborne albedo measurements were made.

## 3 Error sources and albedo correction

There are several sources of error associated with these data. The first is that the instruments had to be in a weighted box rather than mounted to the helicopter due to safety regulations of the helicopter contractor. We believe that the fin and the >400 pounds of ballast in the box maintained a mostly level platform from which the data were collected. This was confirmed in visual observations of all flights from multiple locations in the valley. However, we cannot guarantee that instruments were perfectly level for all measurements, and do not know the exact tilt of the instrumentation for each measurement. Allison et al. (1993) made albedo measurements over Antarctic sea ice by helicopter also with no definite knowledge of the tilt of the instrumentation. They state that tilt affects the upward looking radiometer far more than the downward looking radiometer. We use data from upward looking radiometers located on static, level meteorological stations, and therefore do not need to account for this error. However, this also means that we are not able to use this instrument pair to estimate tilt of the airborne instrument using methods such as those proposed by Weiser et al. (2016). For the downward looking instrument measuring more diffuse reflected radiation, the tilt error is negligible relative to other sources of error, particularly under overcast conditions when reflected radiation is entirely diffuse (Allison et al., 1993). We acknowledge that tilt may introduce some error in our measurements, but is likely only a few percent and far less impactful than the effect of variable sun angles.

Under overcast conditions, the surface properties and orientation of the location of interest do not affect albedo measurements as incoming and reflected radiation is diffuse (Allison et al., 1993; Grenfell et al., 1994; Pirazzini, 2004; Warren et al., 1998). However, under clear skies, there are both direct and diffuse components of incoming solar radiation and therefore the orientation of the sun relative to the surface over which we make measurements can have a large effect on apparent albedo (Grenfell et al., 1994). This is exacerbated at high zenith angles resulting in more oblique radiation at the land surface. This issue has been addressed in Antarctic albedo measurements at the station scale for uniform sloping surfaces (Grenfell et al., 1994) and for sastrugi (Pirazzini, 2004; Warren et al., 1998). Error estimates of apparent albedo in those conditions are generally less than 10%, but are highly dependent on solar zenith angle, slope, and orientation of the sun relative to the surface aspect.

We applied a simple correction factor to the apparent airborne albedo data adapted from Grenfell et al. (1994).

$$\alpha_{true} = \frac{\alpha_{apparent}}{\frac{\cos[\theta_{sun} + \theta_{surf} \ \cos \varphi]}{\cos \theta_{sun}} * [1 - \frac{\theta_{surf}}{2}]}$$  eq. 1

Where $\theta_{sun}$ is the solar zenith angle $\theta_{surf}$ is the slope of the land surface. $\varphi$ is the angle between the sun azimuth and the aspect of the land surface, where 0 is when the sun is directly uphill of the surface. We calculated the slope and aspect of the land surface by calculating the mean slope and aspect of a 100 m radius area centered on the location of the measurement

using a 1m lidar-derived DEM (Fountain et al., 2017). This correction was applied to all points collected under clear sky conditions. Correction factors (denominator of equation 1) varied from 0.14 to 1.19. We found that under relatively high slopes (> 10°) the correction factors became very low, resulting in large increases between apparent and true albedo. Slope is the dominant factor in this correction however $\varphi$, can compound with high slopes to increase the magnitude of this correction factor at very low or high angles. Studies that developed corrections for snow and ice surfaces found that apparent albedo is

generally within 10% of true albedo (e.g. Lhermitte et al., 2014).  In the case of MDV glaciers with high apparent albedo, this correction produced true albedo values much higher than 1 and in some cases as high as 2 (figure 4). This is physically unrealistic and highlights the issue with applying this method to certain surfaces. This correction factor was developed for uniform surfaces of consistent slope and aspect. In the MDV's this method is sufficient for many locations including lake ice, most soil locations, and upper elevations of the glacier ablation zones. The lower elevations of the glaciers are topographically

complex and the surfaces have a wide distribution of slope and aspect. This results in additional scattering, shading, and reflection of light in more complex ways (Wen et al., 2009). Commonwealth Glacier is more uniform and the corrections worked well with data collected over this glacier in all locations (Figure 5d-f). The lower elevations of Canada and Taylor Glaciers have more topographic complexity from the well-developed supraglacial drainage networks, which resulted in very low correction factors and high corrected albedos (Figure 5d-f)

The issue of complex terrain influence on albedo in snow and ice environments has been addressed from the station and satellite scale for a range of features including penitentes, sastrugi, and small rough alpine glaciers with topographic shading (Dumont et al., 2012; Klok et al., 2004; Lhermitte et al., 2014; Warren et al., 1998). Corrections for satellite data can be very complex, and often require knowing the location, field of view, and bidirectional reflectance function for the surface (e.g. Wen et al., 2009). The smaller scale of this airborne albedo data is not well suited to the larger scale satellite correction

methods, due to data requirements and mismatches in the scale over which it is meant to be applied. Albedo corrections at the station scale often require development of an idealized model of the surface and set of assumptions/numerical descriptions of the optical properties of the surface, which is also unrealistic for this application (Lhermitte et al., 2014; Pirazzini, 2004).

Findings of modelling work by Lhermitte et al. (2014) suggest that the higher the sensor is off the surface, it integrates more variability of the surface and lowers the apparent albedo error. They found that a sensor 4m off the surface converges

toward true albedo. We argue that this is true for our data as well, where the position of the sensor 100m above the land surface reduces some error due to high mean slope. While slopes on average are very high, they are of many different aspects and receive and reflect more light than if they were a uniform surface sloping away from the azimuth of the sun. This results in erroneously low correction factors under the simplified method. The irregularity of the MDV glacier ice surface and sediment

cover make correction development a challenging task. More work should be done to develop a correction factor well suited to these conditions, however we believe that this is beyond the scope of this paper and available data. For this reason, we present both the apparent and corrected albedo using the simplified correction proposed by Grenfell et al. (1994; figures 4,5). We present results for soil and lake surfaces using corrected albedos and discuss corrected and uncorrected (apparent) glacier albedo. When comparing across or within glaciers we use the apparent albedo. We believe, based on previous studies and with comparison to station albedos, that glacier apparent albedo is within 10% of true albedo. This is within accuracy required in order to draw conclusions about broad spatial and temporal patterns we observe in the MDVs.

## 4 Results

### 4.1 Seasonal variability of incoming shortwave radiation

During the austral summer, mean daily radiation increases through December as solar elevation increases peaking at the solstice and decreases moving in to the fall. We conducted flights as early as 11 Nov and as late as 23 Jan, in order to capture the seasonal changes in albedo. Therefore, maximum potential incoming radiation varies across flights as a function of day of year. Throughout each season, there is high variability in mean daily radiation due to cloudiness and individual storms (Figure 3a-c). We attempted to schedule flights during completely cloud free days and were successful during the 2016-17 season (Figure 3e). In some cases of persistent cloudiness, we attempted to make flights under uniform cloud cover such as the last two flights of the 2015-16 season. There is limited variability in incoming radiation across the valley on these two days (Figure 3d). Cloud cover is known to increase albedo for snow and ice covered areas and decrease it for other surfaces by several percent (Key et al., 2001). We did not correct albedo on these cloudy days. The goal of this study is to describe broad patterns in albedo rather than exact measures. We believe that the cloudy flights will not influence the main conclusions of this study, but do take it in to account in our interpretation of results.

Canada and Commonwealth glaciers flow out of the Asgard mountains from the north and Lakes Hoare and Fryxell are also on the northern side of the valley (Figure 1). The also valley narrows and steepens toward the Taylor Glacier. Thus, meteorological stations are influenced by topographic shading early and late in the day, particularly by the Asgard Mountains (Figure 3g). We conducted flights between 11 AM and 1 PM to avoid topographic shading, and minimize zenith angle, and variability in azimuth. Zenith angle varied less than 10° across all flights and azimuth ranged across 60°.

Incoming radiation was generally less spatially uniform during the flights in the 2017-18 season (Figure 3f). However, because we are using the closest meteorological station to calculate albedo for a given measurement, this adjusts for some variability in radiation across the valley. Incoming shortwave measured at the Canada Glacier meteorological station during the flight on 27 Dec 2017 averaged 249 W m$^{-2}$ while radiation at other stations ranged between 570 and 651 W m$^{-2}$. We attribute this low incoming radiation to a cloud. Reflected radiation using Canada Glacier station data produced values over 1, suggesting that the cloud was localized and did not shade the entire Canada Glacier. We used an average from the other meteorological stations and the recalculated apparent albedo produced reasonable values. Across the seasons, radiation is

highly variable due to valley-wide weather patterns and patchy cloudiness (Figure 3d-f). These apparent albedo measurements were collected under as uniform conditions as possible and what variability did exist is mostly accounted for by using the closest meteorological station for incoming radiation. This allows us to attribute spatial patterns to true differences in albedo rather than variability in sunniness.

## 4.2 Patterns of albedo across landscape types

We typically collected over 100 points per landscape type resulting in high spatial resolution of albedo measurements synoptically across the entire valley (Table S1). Across landscape types there was distinct separation in the 2015-16 season (Figure 5a, d). There were significant differences in reflected radiation across glaciers, lakes and soils. Glaciers consistently had the highest, and soils the lowest albedo (p-values all <<0.05). This follows expectations as it can be observed visually that glaciers, particularly MDV glaciers, have very white surfaces. Lakes, while ice covered, tend to have darker surfaces than glaciers due to aeolian sediment deposition. Soils naturally have low albedo due to their rocky nature. We also found low variability in soil albedo across all flights in the 2015-16 season. This is also expected as soils are generally mixed glacial till and have no vegetation cover. Glaciers and lakes typically have much higher variability. Due to the 100 m buffer around the lake edges, any change in moat condition is not reflected in these measurements (i.e. no open water is included in these measurements). Rather, the variability in and change of lake albedo is due to sediment cover and structure of the lake ice. Glaciers also have variable surface albedo. Taylor and Canada Glacier ablation zones have flat ice at higher elevation with centimeter-scale roughness, and large supraglacial drainage networks at low elevation with meter-scale roughness, patches of aeolian and colluvial debris, and large ice-covered melt ponds. Commonwealth Glacier does not have a large drainage system but does have supraglacial streams and meter scale cryoconite holes dispersed across the flat ice surface. The surface structure of glaciers and lakes result in variable melt and ablation as well as the ability to trap windblown material, such as sediment and snow.

## 4.3  Along valley gradients in albedo

In order to understand how albedo varies along a transect from west to east along the valley with decreasing distance to the ocean, we plotted albedo mean and standard deviations of each of the three main glaciers and lakes (Figure 5). Soil data are divided into west (Bonney and Hoare basins) and east (Fryxell basin) parts of the valley. We also plot measurable snow events observed at Lakes Bonney, Hoare, and Fryxell meteorological stations as snow 'hyetographs'. The snow events, particularly in the 2016-17 season, have a dramatic effect on the valley wide albedo patterns.

The first flights of the 2015-16 and 2016-17 seasons had similar albedo patterns. All glaciers and lakes have the lowest albedo measured all season (Figure 5a, b). Lake Fryxell has significantly higher albedo than Lakes Hoare and Bonney. On the second flight of the 2015-16 season with no measured snowfall between flights, we observe that albedo of all lakes increased and there is no difference between the lakes (Figure 5d). Albedo of the glaciers did not change significantly.

On November 22$^{nd}$ and 23$^{rd}$ 2016, a very large, valley-wide snow event occurred. The greatest accumulation was observed at the Lake Fryxell meteorological station. Accumulation decreased inland toward the Bonney basin (Figure 5b) and persisted for several weeks. The snow event is evident in albedo measured during the subsequent two flights. The longitudinal increase in snowfall with proximity to the coast coincides with increased albedo moving east toward the coast. There is
increasing albedo from Taylor to Canada to Commonwealth glaciers (Figure 5b). Lake Fryxell has the highest measured albedo, even higher than all three glaciers. Similarly, eastern soils have higher albedo than Lakes Bonney, Hoare and the Taylor Glacier. Mean albedo over eastern soils on 03 Dec was 0.60, indicative of substantial snow cover. Both east and west soil albedo standard deviation increased in the two flights after the snow event, but did not markedly increase for the glaciers or lakes. Eleven days after the snow event, we still observe an east-west pattern of albedo. Lake Fryxell had the highest albedo
of any feature. While albedo of eastern soils decreased slightly, mean albedo remained at 0.49, which was still higher than both Lakes Hoare and Bonney. The albedo of Canada and Commonwealth Glaciers actually increased (Figure 5b). Finally, albedo of Lake Fryxell remained elevated for the rest of the season relative to albedo measured over Lake Fryxell at the end of the previous snow-free season (mean corrected albedo was 0.54 and 0.58 vs 0.46 and 0.48 on the last two flights of the 2016-17 and 2015-16 seasons respectively).

One of the most prominent differences in longitudinal patterns of albedo was measured on 22 Nov 2017. We observed an along valley increase from west to east in albedo on that flight similar to what was observed after the snow event the previous season described above. The last snow event recorded prior to the 22 Nov 2017 flight was in August. At the beginning of the 2017-18 season there were very large snowdrifts across the valley against the toes of glaciers, in incised stream channels, and in the lee of camp buildings. We found on this first flight that while western soils did not have elevated albedo, eastern
soils have a mean albedo of 0.64 (Figure 5f). Commonwealth glacier also had the highest mean albedo observed on any flight across all three seasons (0.74, Figure 5). By early December 2017, the albedo of all glaciers, lakes and soils had decreased. Several small snow events occurred in December but they did not have a clear spatial effect like what was observed in the previous season. The last two flights of the 2017-18 season did have elevated albedo on glaciers and lakes relative to the previous two seasons, possibly attributable to snow and cold temperatures restricting melt (Figure 6). Overall, the three seasons
provide contrast in spatial and temporal patterns of albedo due to the weather that occurred during each. 2015-16 was a clear and relatively snow free season. 2016-17 had a very large and persistent snow event in early season. 2017-18 started with large snow drifts from winter snow and subsequent wind redistribution. This facilitates discussion of how the landscape responds to each type of albedo-altering event.

## 4.4 Glaciers have spatially and temporally variable albedo

We hypothesized that there would be a positive relationship between albedo and ice surface elevation. Following standard principles of adiabatic cooling, more melt will be generated at lower elevations, lowering albedo due to the presence of liquid water that creates melt pools, exposes and concentrates sediment, and creates increased surface roughness (Bagshaw et al., 2010; Hoffman et al., 2016). We compare apparent albedo to ice surface elevation of each measurement collected on

each flight and season over Taylor (Figure 7) Canada (Figure 8) and Commonwealth Glaciers (Figure 9). We also plot hypsometric curves for each glacier to compare how much glacier area is represented by the measurements. Albedo measured at the meteorological station on that glacier at the time of the flight is also plotted as the larger colored point in each plot.

On almost all flights, we have a measurement directly above the meteorological station. On the third flight of the 2016-17 season, the pilot turned around over Lake Bonney and we do not have measurements of albedo over the Taylor Glacier. The airborne measurements collected at the elevation of the meteorological station are comparable to the station-based measurements for all flights over Taylor Glacier giving us confidence in the use of apparent albedo for these comparisons. On the flights with little variation in albedo, the ground-based measurement is fairly representative of the entire ablation zone albedo. However, over Taylor Glacier we observed an albedo increase with increasing elevation on several flights, particularly the second flight of the 2017-18 season (Figure 7k). Additionally, at lower elevations we observe a higher apparent albedo on one flight line relative to the other, with a difference as much as 0.10 on some flights (Figure 7). There tends to be a convergence of albedo between the two flight lines at around 300m elevation. Taylor Glacier has lower variability in apparent albedo between flight lines and from flight to flight than Canada and Commonwealth Glaciers.

Canada and Commonwealth Glaciers have very clear separation between the two flight paths across the glaciers (Figure 8, 9). On Commonwealth Glacier, there is very distinct separation between the upper and lower flight lines by elevation. The lower flight line rarely reaches the elevation of the upper flight line (Figure 9). The shape of both Canada and Commonwealth Glaciers is such that on a transect across the ablation zone, perpendicular to glacial flow, the highest elevation is in the center and lowest on the edges (Figure 8 and 9). We find that for a given elevation there is a higher albedo on one side of the glacier relative to the same elevation on the other side (Figure 8b). We plot the locations of measurements taken over each glacier with their color representing albedo (Figures 10-12). There is a clear longitudinal difference in albedo on the Canada and Commonwealth Glaciers on most flights. We believe these differences are larger than what would be caused by differences in aspect and solar illumination across the two sides. The western side of both glaciers tend to have lower albedo than the eastern sides, and is also visible in the imagery. Figures 11n and 12h are good examples of this pattern. The last several measurements collected on the eastern edge of the Commonwealth glacier decrease slightly. The darker edge of the glacier is visible in the true color imagery (Figure 12).

While the station-based measurement of albedo is more frequently representative of Taylor Glacier (Figure 7), it is typically near the highest airborne albedo measured over Commonwealth and Canada Glaciers (Figure 8 and 9). The meteorological stations on these glaciers are both located in the center and toward the higher elevations of the ablation zones (Figure 1,8,9). According to our airborne measurements, we generally see an increase in albedo with increasing elevation and from west to east across the ablation zones. Therefore, these meteorological stations give us confidence in using apparent albedo but, due to their siting, are not particularly representative of the albedo of the entire Canada and Commonwealth Glacier ablation zones.

We compared MODIS albedo data to both apparent and corrected albedo (Figure 13). We found that overall fits between the MODIS data and airborne data are better using apparent (uncorrected albedo). This increase in root mean square

error using corrected data is likely due to the unrealistically high values of corrected glacier measurements. In the uncorrected data, most soil points fell below the 1:1 line (Figure 13a). The corrected data cluster centered around the 1:1 line, suggesting the correction worked well for soils and, while there is a lot of variability, there is good agreement between corrected soil albedo and MODIS albedo. Lake albedo values generally fell above the 1:1 line in both the corrected and uncorrected datasets (Figure 13). Because lakes are flat, the corrections do not substantially alter the lake albedo values. Glacier measurements actually cluster around the 1:1 line in the uncorrected data, suggesting that the uncorrected glacier data does reasonably well in representing true albedo. The corrected glacier data pulls many measurements much higher, and many more glacier values are far above the 1:1 line, leveraging the slope of the best fit line. MODIS data compare relatively well to airborne albedo measurements. The spread of data relates to the mismatch of scales. MODIS is integrating across a much larger area than the airborne measurements.

## 5 Discussion

### 5.1 Seasonal albedo shifts in the absence of snow

Snowfall alters the energy balance of a surface because it has different thermal and optical properties than the underlying ice. Therefore, one would expect there to be an effect on the seasonal aging of ice under even minimal snow cover of moderate duration. There was almost no measurable snow in the first season in which we conducted this study. This allows us to make observations of albedo seasonal change due to ice aging without the influence of snow over both lake and glacial ice. We found that in the 2015-16 season, glacier and lake ice albedo increased from the first flight in late November to the third flight in late December (Figure 5). This albedo increase is what some refer to as ice whitening and is a result of ice structural change which can increase scattering of light (Fritsen and Priscu, 1999; Howard-Williams et al., 1988). The changing grain/crystal shape is a function of the initial structure and the surface energy balance (Adams et al., 1998; Grenfell, 1983; Warren, 1982; Wiscombe and Warren, 1980). During the period of albedo increase we observed over the MDV lakes and glaciers from November to late December, total mean daily solar radiation and air temperature increased, which resulted in more energy available to warm, melt, and alter the ice surfaces (Figure 3a, 6a). The exact mechanisms of ice whitening differ on lake and glacier surfaces due to the differences in how the ice was formed and processes governing melt.

The process of ice structural change and its effect on albedo and the transmission of light has been well studied on MDV lakes. McKay et al. (1994) developed a conceptual model for the seasonal evolution of Lake Hoare ice cover. This model suggests that early in the season the ice optical properties are nearly constant. However as solar radiation and temperature increase, albedo also increases due to fracturing, internal melting, and the development of light-scattering Tyndall structures (Howard-Williams et al., 1988; Knight and Knight, 1972). This increased albedo coupled with decreased transmission of light peaks in mid-summer at the peak of incoming solar radiation (Figure 3,4,5; McKay et al., 1994). We observe that lake ice albedo decreased in the last two flights of the 2015-16 season (Figure 5d). By mid-summer, enough melt has occurred that the light-scattering melt structures are destroyed as an aquifer develops within the ice cover (Fritsen and Priscu, 1999; Howard-

Williams et al., 1988; McKay et al., 1994). Eventually as solar radiation and air temperatures begin to decrease, the ice will refreeze, redevelop structure, and increase scattering and albedo (Figure 3,6). We believe that we did not have a flight late enough in the season to capture this process (which is in part a limitation of our field access, ending in late January each austral summer).

5       The mean lake ice albedo we measured in the 2015-16 season follows the conceptual model of McKay et al. (1994). However, melting and surface optical properties of lake ice are very complex due to meter-scale surface roughness and patchy surface characteristics. Sediment concentration on lake ice ranges from 0.2 to 2 g cm$^{-2}$, which at highest concentration is essentially a sand dune with ice filling the pore space (Wharton et al., 1989). These high sediment concentration patches can be several meters in diameter, melt down to 10's of centimeters below the surrounding ice surface, and hold ponds of liquid

melt water. The adjacent ice surface is much whiter and is typically characterized by tables of candle ice. This variability in surface characteristics can explain some of the variability we observe in our albedo measurements (Figure 4b, 5d). Furthermore, Lakes Hoare and Fryxell have more pronounced sediment pond- candle ice patchiness than Lake Bonney, which may be a factor in why we observe different temporal dynamics and generally lower standard deviations of albedo on Lake Bonney (Figure 5d).

15       The ice structure and melt processes are very different for the MDV glaciers vs. lakes. The energy balance dynamics generating melt have been extensively modeled (Hoffman et al., 2014, 2016; Lewis et al., 1998), but the result of melt on the optical properties of the ice have not been well described. Hoffman et al. (2014) found that melt is generated in the shallow sub-surface of the glaciers (5-10 cm below the surface) and drains to topographic lows, reducing the ice density. As ice density is lost, light can be scattered in the interstices of the ice grains and increase albedo (Grenfell, 1983; Grenfell and Maykut,

1977). Another potential mechanism for albedo increase is due to sublimation (Winkler et al., 2009). Sublimation makes up a high proportion of total ablation on the glaciers and is by far the dominant mechanism of ablation on the glacier surface relative to melt (Lewis et al., 1998; Hoffman et al., 2014). However, there are mixed effects of sublimation (Winkler et al., 2009). Sublimation can reduce ice density on daily timescales and create more microtopography which would increase scattering and can change on sub-daily timescales (Pirazzini, 2004). Conversely, sublimation-derived microtopography could increase the

ability for the glacier to trap sediment, and scatter light back into neighboring ice, thereby reducing albedo (Lhermitte et al., 2014; Winkler et al., 2009). This effect is likely spatially variable depending on the degree of sublimation and melt in a given location, would require more extensive modelling, and is the subject of future work.

       Seasonal increase in ice albedo has been observed on glaciers elsewhere including temperate glaciers in the Alps and polythermal glaciers in the Arctic (Knap and Oerlemans, 1996; Oerlemans and Knap, 1998). However, this albedo increase

has largely been attributed to the formation of superimposed ice, where meltwater derived from snow refreezes upon reaching colder ice. The MDV glaciers generally do not develop a seasonal snow cover in winter so superimposed ice cannot form. However, the principle of refreezing meltwater is probably still applicable on these glaciers (Hoffman et al., 2016). While some meltwater generated on the glaciers drains through the ice matrix into supraglacial channels and leaves the glacier as

terminal waterfalls, some meltwater will refreeze before it can leave the glacier and likely contributes to the increased albedo. Temperatures generally remain below 0 throughout the seasons, promoting refreeze (Figure 6d-f).

The glaciers had a slight decrease in albedo in the latter half of the season, particularly on Canada and Commonwealth Glaciers (Figure 5a). The flight lines were not as consistent on Taylor Glacier, making it hard to compare and understand temporal shifts in albedo on this glacier. In the last two flights on both Canada and Commonwealth Glaciers the flight line traversing upper elevations had the largest albedo decrease while measurements made over lower elevations were more constant (Figures 11 and 12c-e), also shown the in the albedo timeseries measured at the meteorological stations (Figure 6a-c). The lower elevations of the glaciers have higher surface roughness and slope, and are drained by supraglacial channels. These lower sections may have more stable albedo because they can generate more meltwater, effectively drain it, and are not as affected by melt-refreeze processes (Hoffman et al., 2016). Higher elevation glacier surface is characterized by smooth ice with cryoconite holes. Cryoconite holes are locations where wind deposited sediment has concentrated and melted in to the ice forming a cylindrical depression with a ~ centimeter thick layer of sediment on the bottom, overlain by several centimeters of liquid water (Fountain et al., 2004). Antarctic cryoconite holes tend to have ice lids that by late season can melt, leaving open holes where the light-transmitting liquid water and darker sediment is exposed to the atmosphere (Fountain et al., 2004, 2008). This late season opening of cryoconite holes driven by a short period of warm temperatures (Figure 6d) may be the cause of the reduced albedo we observed in the 2015-16 season. We suggest that seasonal trends in glacier and lake ice albedo in the absence of snow are a function of the seasonal energy balance and the physical ice structure from the grain to entire lake/glacier scale.

Soil albedo patterns are much different than lakes and glaciers due to their fundamental difference in material: rock vs. ice. In 2015-16, soil albedo decreases slightly from late November to late December (mean albedo: 0.18 to 0.12, Figure 5d) possibly due to melt of residual snow patches. Soils can have decreased albedo when wet (Bøggild et al., 2010). In addition to snowmelt, hypersaline patches of soil MDVs can become wet due to deliquescence and may contribute to soil albedo decrease over the season without a snow event (Levy et al., 2012). There is strong variability in soil albedo patterns across seasons (due to snow), but across all seasons soil albedo is lowest at the end of the season (Figure 4c).

**5.2 Snow alters spatial patterns of surface albedo**

Snow typically has a higher albedo than ice and therefore snowfall in the Taylor Valley raises the albedo of every landscape feature. The albedo of snow is strongly influenced by grain size, snow age, and history; and in addition, snow must be of sufficient water equivalent depth for a surface albedo to be set by the snow alone (Wiscombe and Warren, 1980). Snow events are small in the MDVs, only producing a few mm of water equivalent in a given event, typically follow a gradient of increased snowfall toward the mouth of the valley, and is generally quickly redistributed by wind (Doran et al., 2002; Fountain et al., 2010). We broadly discuss the role of snow in altering the albedo of each landscape feature and discuss some possible mechanisms for the patterns we observe.

On 22-23 Nov 2016, the Taylor Valley received one of the largest snow events on record, with the highest recorded snowfall toward the valley mouth in the Fryxell basin (Figure 5b). This spatial gradient in snow fall was reflected in the albedo of the landscape features. Taylor Glacier, Lake Bonney and soils in that basin only saw modest increases in albedo. In addition, the Bonney and Hoare basin soils had increased standard deviation (Figure 5e). It is possible that the underlying soil surface may be contributing an albedo signal mixed with the thin snow cover, which is variable depending on the soil surface structure (distribution of coarse fragments and boulders) as well as possible variability in snow depth due to wind redistribution. Lower in the valley, there is a much larger increase in albedo after the snow event (Figure 5e). Again, this may be both a function of the initial accumulation as well as potential wind redistribution. The strongest winds tend to move off the polar plateau along the valley (Doran et al., 2002; Nylen et al., 2004), moving snow from west to east. These increased snow depths likely cover the lower valley more completely, reducing surface roughness and masking the signal of low albedo underlying surfaces, resulting in a larger increase in albedo (Warren et al., 1998).

Typically, as snow ages grain size increases, impurities accumulate (dust and black carbon), and albedo decreases. This has been shown empirically and has been used in models across scales to reasonably reproduce measured albedo timeseries (Oerlemans and Knap, 1998; Pellicciotti et al., 2008; Warren and Wiscombe, 1980; Wiscombe and Warren, 1980). We hypothesized following this principle that albedo would decrease across the landscape in subsequent flights after a snow event. Surprisingly, we found mixed results where the soils mean albedo significantly decreased, and Lake Hoare decreased slightly but not significantly. Canada and Commonwealth Glaciers and Lake Bonney increased significantly, while Lake Fryxell mean albedo increased but not significantly (Figure 5b, e). There was a period of high solar radiation and two short spikes in air temperature between the flights that may have promoted melt or sublimation particularly for the soils with thinner snow cover (Figure 3b, 6e).

Variability in the albedo change between lakes could be due to a combination of the gradient in snow accumulation and sediment and snow capture efficiency of the lake ice. Lake Bonney historically has the lowest sediment load (Sabacka et al., 2012) and receives the least amount of snow (Fountain et al., 2010) leaving more exposed ice cover that is subject to ice-whitening independent of the snow event. Lake Hoare typically has the greatest sediment load (Sabacka et al., 2012), which promotes snow melt and once exposed has lower albedo. The increase in albedo at Lake Fryxell could be from snow deposited on the lake surface from wind.

We compared the individual measurements on Canada and Commonwealth Glaciers to understand if there is a spatial pattern of albedo on the glaciers post-snowfall. We found that the lateral difference in albedo for a given elevation is very high (Figures 8h and 9h). The west sides have lower albedo than the east sides of both glaciers on most flights, however the difference became more pronounced for the 03-14 Dec 2016 flights (Figures11g, h and 12g, h). The Canada and Commonwealth Glaciers both flow to the valley bottom and present major topographic obstacles to wind (Figure 1). It is likely that any wind event that occurred between the snow event and subsequent flights transported snow from the valley floor and windward side of the glaciers and deposited them on the leeward side of the glaciers. We believe that the aeolian redistribution

of snow enhanced the spatial gradients in albedo across the glaciers and raised mean albedo. The lakes are not a topographic obstacle to wind and this may be why there was a less pronounced change over the lake ice surfaces.

Aeolian transport of material is one of the most important drivers of connectivity across the Dry Valley landscape and contributes to the morphological and energy balance characteristics for a given landscape position (Sabacka et al., 2012). We found aeolian transport is not only important for driving snow redistribution on the event scale but also may drive patterns of albedo at the seasonal scale. On the first flight of the 2017-18 season we observed a large difference in albedo of the Fryxell basin relative to Bonney and Hoare basin lakes, glaciers, and soils (Figure 5f). Taylor and Canada Glaciers as well as Lakes Hoare and Bonney and associated soils all had albedo comparable to the first flights of the previous two seasons, although there is a high standard deviation measured over Lake Bonney. In the Fryxell basin, Commonwealth Glacier had the highest mean albedo, and both Commonwealth Glacier and Lake Fryxell had the highest standard deviation measured across all seasons. The last measurable snow event occurred in August, 2017. Upon arriving in the valley at the beginning of the 2017-18 season, we found many meter-scale snowdrifts in any location that was a windbreak (i.e. incised streambeds, leeward sides of buildings, terminal glacier fronts), which persisted long in to the summer.

During the winter, Fountain et al. (2010) suggest that due both to the gradient of decreasing snowfall with distance from the coast as well as the foehn wind driven along valley transport and redistribution of snow from valley walls to floors, snow does not persist for very long in the Bonney basin but can settle in persistent drifts in the Fryxell basin. Our albedo measurements agree with their observations. The high standard deviations we observe are likely from the drifted snow, where there are large drifts of high albedo snow in topographic lows interspersed with patches of lower albedo bare ice and soil. Longitudinal patterns of increasing albedo from west to east existed on the Commonwealth Glacier, but the trend was not as smooth as observed on other flights (Figure 12k). In addition, the lowest observed albedo of that flight was near the meteorological station which is at high elevation with the smoothest ice and therefore is least likely to hold snow in high winds (Figure 9k). Interestingly, all components of the Fryxell basin had albedo comparable to the rest of the valley by the next flight on 27 Dec 2017. There was a clear, sunny period between the two flights that may have promoted melting and sublimation of the snowdrifts (Figure 3c). Our findings demonstrate that snow can have a strong influence on albedo across the landscape. We also suggest that redistribution via wind and melting or sublimation of snow can result in large spatial and temporal variability of these patterns. This has been found across the Antarctic continent (Pirazzini, 2004).

**5.3 How representative are apparent albedo measurements?**

Albedo is one of the most important parameters for glacial energy balance across all types of glaciers at all latitudes (Brock et al., 2000; Brun et al., 2015; Dumont et al., 2011; Mölg and Hardy, 2004; Pellicciotti et al., 2008). Remotely sensed observations have been used successfully in many parts of the world, particularly at large scales (e.g. Knap & Oerlemans, 1996; Pope & Rees, 2014). For the McMurdo Dry Valley glaciers, satellite imagery is particularly challenging to use effectively because the oblique angles of most images, topographic shading, and strong atmospheric interference. Additionally, the glaciers are only several km across so the resolution of most images is not sufficient to capture a pure glacier pixel that is

actually representative. Use of satellites to calculate glacial energy balance components in the dry valley region has been limited to a few studies (Bliss et al., 2011; Dana et al., 2002). Largely, albedo in glacial energy balance models has been derived from meteorological stations (Hoffman et al., 2008, 2014, 2016). We compare airborne albedo measurements to both station and remotely sensed (MODIS) data collected during the same periods.

5       The meteorological stations were sited at high elevation in the ablation zone of the glaciers and on horizontal ice with low surface roughness. Hoffman et al. (2014) suggest that the location of these radiometers may not measure the most representative albedo for the entire glacier surface and cause mass balance models to under-predict glacier ablation. We believe that the helicopter-based measurements of albedo are comparable to meteorological station measurements and therefore allow us to evaluate how albedo measured at a station compares to the distribution of albedo across the ablation zone of an entire

glacier (Figure 7-9). With very few exceptions, the airborne measurements return the same albedo as the station-based measurements when directly over the meteorological station. However, during most flights the station-based measurements were not particularly representative due to spatial trends in albedo (Figures 7-9). The airborne albedo measurements over glaciers revealed three key findings: 1) There is generally a positive relationship between albedo and elevation however, 2) for a given elevation there can be a gradient in albedo laterally across a glacier and 3) the strength of these trends can vary

through time, presumably with other energy balance components such as solar radiation, temperature, and wind speed (Figures 3-6).

        We suggest adiabatic lapse and the change in ice structure from drainage-dominated to flat smooth ice resulted in the frequently observed positive relationship between albedo and elevation, common across all three of our study glaciers (e.g. Figures 7k, 8d, and 9i). We also observe across all three glaciers that for a given elevation there can be a major disparity in

measured albedos. On the Taylor glacier, major drainage channels begin to develop roughly 3.5 km from the terminus, 300 m a.s.l., and drain to the terminus parallel to glacier flow. These channels can be tens of meters wide and deep near the terminus and hold darker melt pools and thick layers of sediment. We believe that the difference in albedo we observe for a given elevation on Taylor Glacier depends on if the instrument was over a channel or smooth ice ridge between drainages.

        For the Canada and Commonwealth Glaciers, different albedo for a given elevation is due to lateral differences in

albedo where the west sides tend to be darker than the east sides. The west sides of these glaciers are the upwind side for foehn events that can deposit up to 1 kg m$^{-2}$ yr$^{-1}$ on the glacier surfaces and scour snow and ice (Lancaster, 2002; Sabacka et al., 2012). This may be a positive feedback mechanism where the sediment deposition lowers albedo, promotes melt and the development of drainage networks, and increases surface roughness, further lowering albedo. These differences in albedo across the glaciers are not adequately captured by single measurements at the meteorological stations but appear to be related

to morphology, snow, and wind events.

        MODIS data provide more confidence that the airborne measurements are representative of true albedo over these surface types. The corrected soil and lake data agree well with MODIS-derived albedo data. Apparent (uncorrected) glacier data agrees better with MODIS albedo data than corrected data, due to complexities described in section 3. MODIS pixels are 500m resolution. This means that in individual value is integrating over a much larger area and has issues with mixed pixels,

particularly toward narrow parts of the features (e.g. upper elevations of Canada and Commonwealth Glacier ablation zones, Figure 1). There are many induvial airborne measurements contained within the same MODIS pixel. As shown by the vertical linear striped appearance of the data in Figure 13, there is a range of airborne values for any given MODIS value. This illustrates the utility of the airborne data in that it is capable of picking up the smaller scale spatial variability of the land surface. Airborne

measurements also do not have issues mixed pixels like MODIS does. Lakes and glaciers would have artificially low values if part of the pixel also covers soil and vice versa for soils having artificially high albedo. The best relationships between MODIS and airborne data are following snow events (Figure S1,2). The broad valley-wide patterns in snow distribution result in clearer albedo differences at both measurement scales. As snow melts, soil and sediment patches are exposed, ice ages, and relationships deteriorate. This mismatch in measurement scale is apparent here as patchiness increases and albedo spatial

variability increases. More work should be done to develop a predictive model of glacier albedo capitalizing on the strengths of all three measurement scales (station, airborne, and MODIS) and incorporate it in to spatially distributed energy balance models in order to better predict ablation and meltwater generation from these glaciers.

## 6  Conclusions and implications for valley-wide connectivity

15        The transfer of energy and water among landscape units is critical for sustaining life in this polar desert. The glaciers supply melt water to the downstream environment, the generation of which is determined by the energy balance. Albedo is a key parameter in the energy balance equation for all glaciers, but particularly in this landscape where melt is primarily driven by solar radiation rather than sensible heat. The amount of energy that is absorbed or reflected also has important ecological implications across the landscape from the under-ice primary productivity in lakes to the thermal stability of soils for

invertebrate populations (Fritsen and Priscu, 1999; Wlostowski et al., 2018). While we have the benefit of many meteorological stations dispersed throughout the valley, only those located on glaciers measure albedo, and they are limited in that they are point measurements in a highly spatially variable landscape.

        The meteorological stations appear to sufficiently capture variability in incoming radiation, which on a seasonal basis varies with cloudiness (Figure 3) and varies on annual time scales with atmospheric concentrations of pollutants (Obryk et al.,

2018). The amount of reflected radiation appears to vary seasonally with the amount of incoming radiation, snow events, wind events, and the physical characteristics of the landscape. We suggest that wind driven snow and sediment redistribution is a critical factor in the long- and short-term spatial patterns of albedo that we observed: both broadly increasing albedo from west to east along valley and individually across the Canada and Commonwealth glaciers. Wind drives connectivity across the landscape and is a feedback mechanism for the energy balance of a given landscape unit. By delivering sediment and

redistributing snow, it promotes meltwater generation and drives further connectivity. These longitudinal patterns in albedo indicate that point measurements made at a meteorological station are not sufficient to fully describe the albedo of a given ice or soil environment and that one can expect a very different energy budget from one basin to the next. This study is a step toward understanding how we can scale our current measurements to the entire valley and the factors we would need to consider in order to do so. Further development of this dataset, methods for correcting and coupling it with other albedo data

(i.e. meteorological stations, and remotely sensed data) and modeling work, particularly incorporating redistribution of snow and variability in surface roughness, should be done to understand the impacts of albedo variability on the meltwater generation from glaciers as well as the energy budget for lake ice and soils.

## Author Contribution

Bergstrom and Gooseff designed the study and carried out data collection. Bergstrom performed data analysis. Myers contributed snow data. Cross contributed MODIS data analysis. Bergstrom prepared the manuscript with contributions from all co-authors.

## Competing Interests

The authors declare that they have no conflict of interest.

## Data Availability

Meteorological and albedo datasets are available on MCM LTER database: mcmlter.org.

## Acknowledgements

The authors would like to acknowledge the civilian contractors and Petroleum Helicopters, Inc. who have supported the US Antarctic Program to make this data collection possible. This research is a part of the McMurdo LTER, funded by the National Science Foundation grants 1115245 and 1637708. Support was also provided by a National Science Foundation Graduate Research Fellowship awarded to Bergstrom. Any opinions, finding, conclusions, or recommendations expressed in the material
are those of the authors and do not necessarily reflect the views of the National Science Foundation.

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

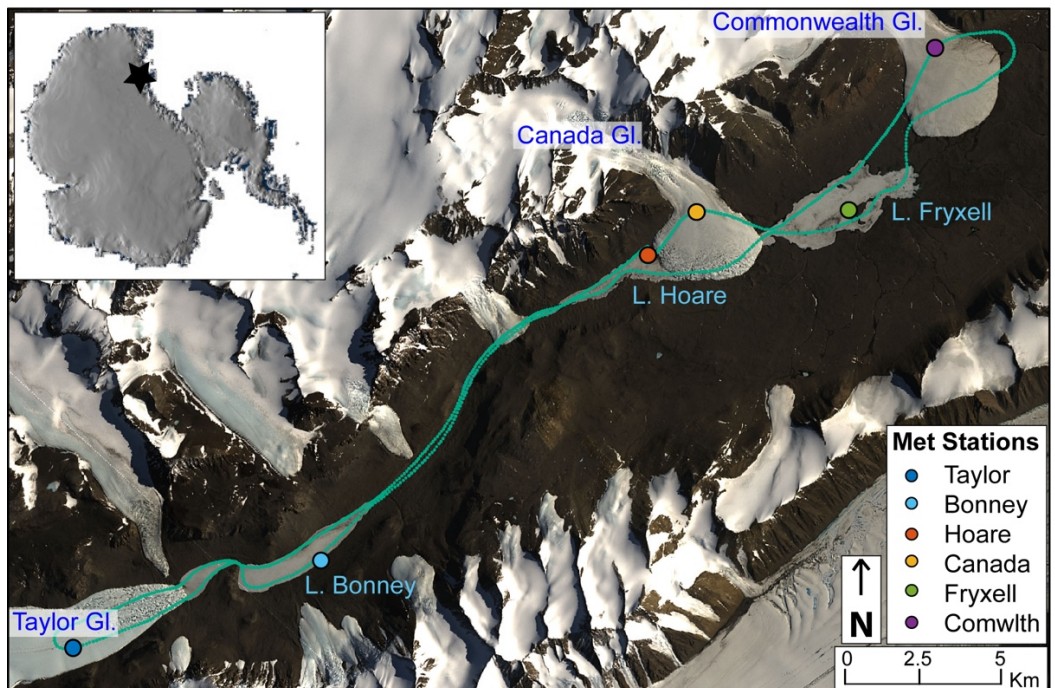

Figure 1. The Taylor Valley, Antarctica. This study focuses on the three labeled glaciers (Taylor, Canada, and Commonwealth) and lakes (Bonney, Hoare, and Fryxell). Meteorological stations measuring incoming and reflected shortwave radiation are shown as points. An example flight path, flown January 12th, 2016, is shown by small green points. Background imagery is provided by USGS/NASA; Landsat8 image collected on 14 January, 2016.

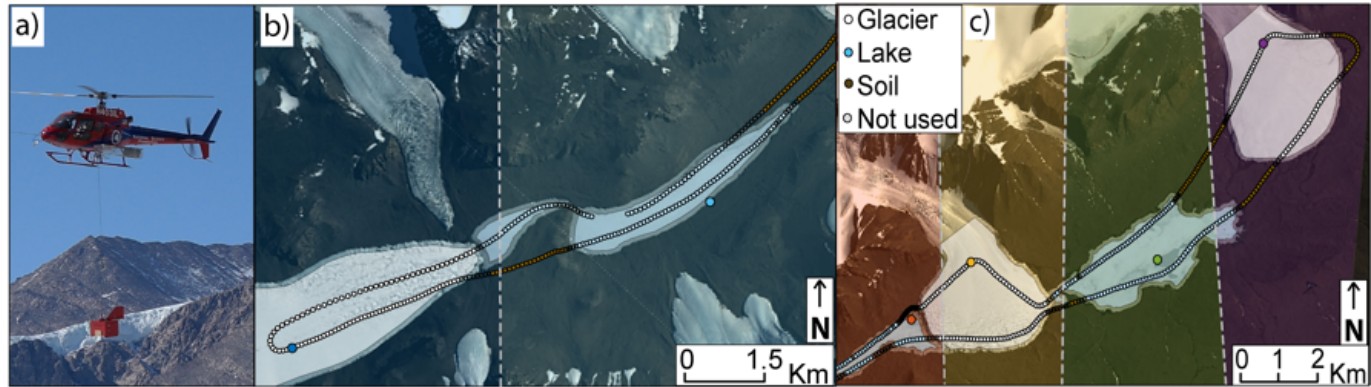

Figure 2. Surface reflectance data are collected using a radiometer mounted underneath a weighted box and flown by helicopter across the valley (a). All flights are processed to associate each measurement with a specific terrain feature and the closest meteorological station. An example of a processed flight line over Taylor Glacier and Lake Bonney is shown in b) and over Lakes Hoare and Fryxell and Canada and Commonwealth Glaciers in c), flown January 12th 2016. In b) and c) white points are assigned to glaciers, blue points are assigned to lakes, brown to soils, and open circles were data that were not used due to proximity with the edge of a feature (<100 m). Polygons of the buffers used for glacier and lake edges are also shown. Thiessen polygons were drawn to find the closest meteorological station for each measurement, shown as a polygon of the same color of the meteorological station point. Background imagery is provided by USGS/NASA; Landsat8 image collected on 14 January, 2016.

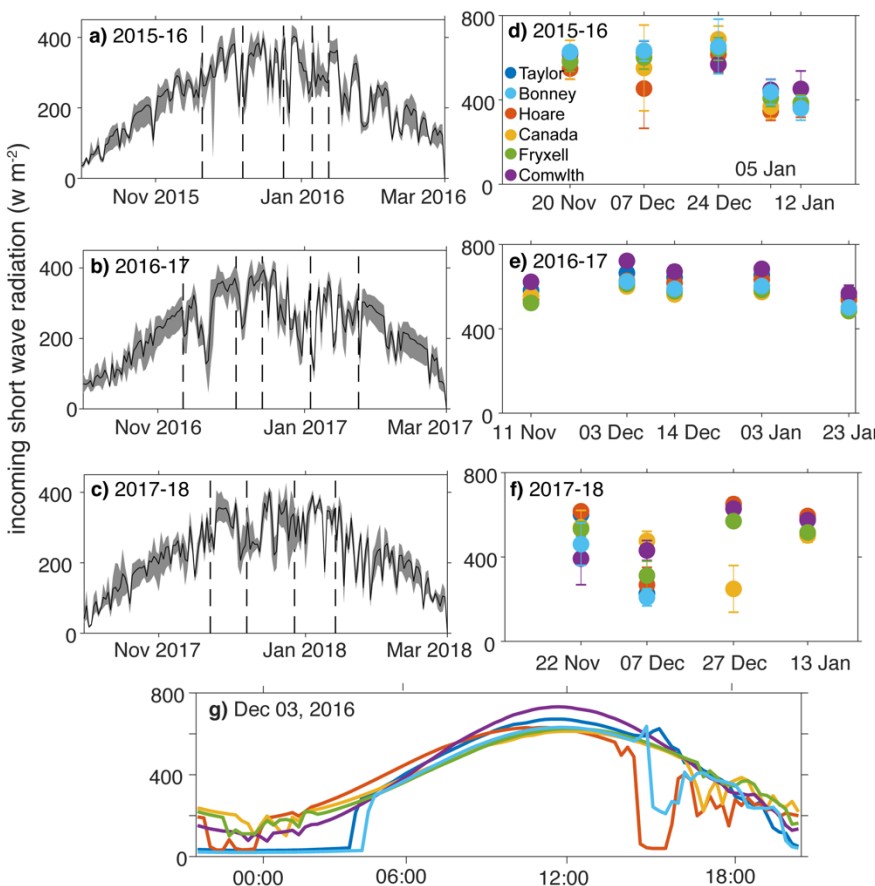

Figure 3. Daily incoming radiation averaged across all met stations (black line) and minimum and maximum radiation (grey shading) from October 1st to March 1st are shown for the a) 2015-16, b) 2016-17, and c) 2017-18 seasons (Doran and Fountain, 2019a, 2019b, 2019c, 2019d, 2019e, 2019f). Mean incoming shortwave radiation measured over a 2-hour period covering the flights conducted during each of the three seasons with error bars indicating one standard deviation (d-f). All three plots are over the same period (November 5th to January 25th) with tick marks at days of flights. Typical daily incoming shortwave radiation at all six meteorological stations is shown in g), local solar time. Periods of low radiation at the Bonney, Taylor, and Hoare stations are due to topographic shading.

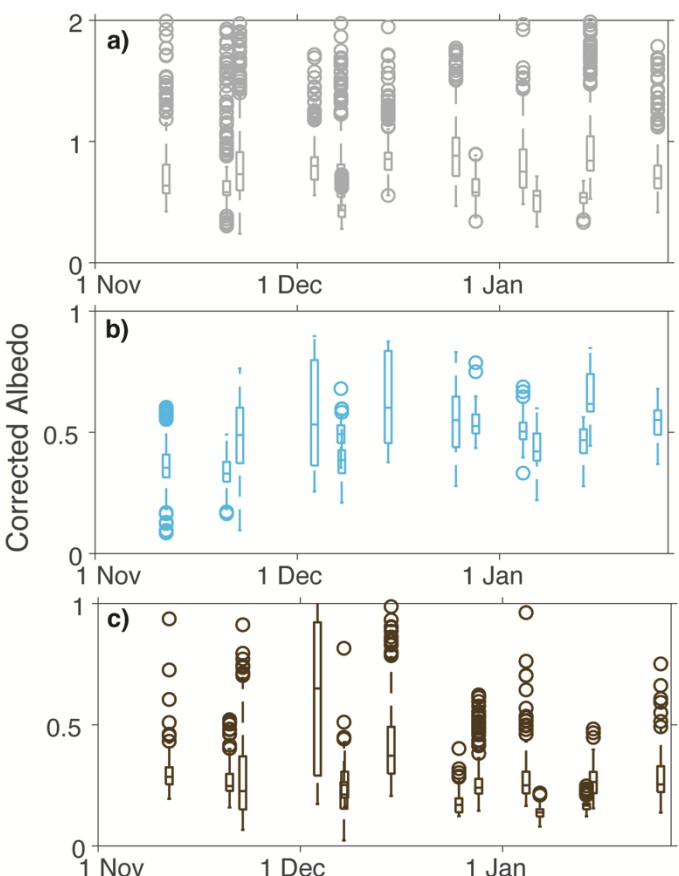

Figure 4. Box plots of corrected albedo from all points measured over a) glaciers, b) lakes, and c) soils. All flights across seasons are shown in plots of each surface type. Boxes are one standard deviation with a line at the mean. Whiskers are 99 percent of data with open circles signifying outliers. Note different y-axis scale for glaciers.

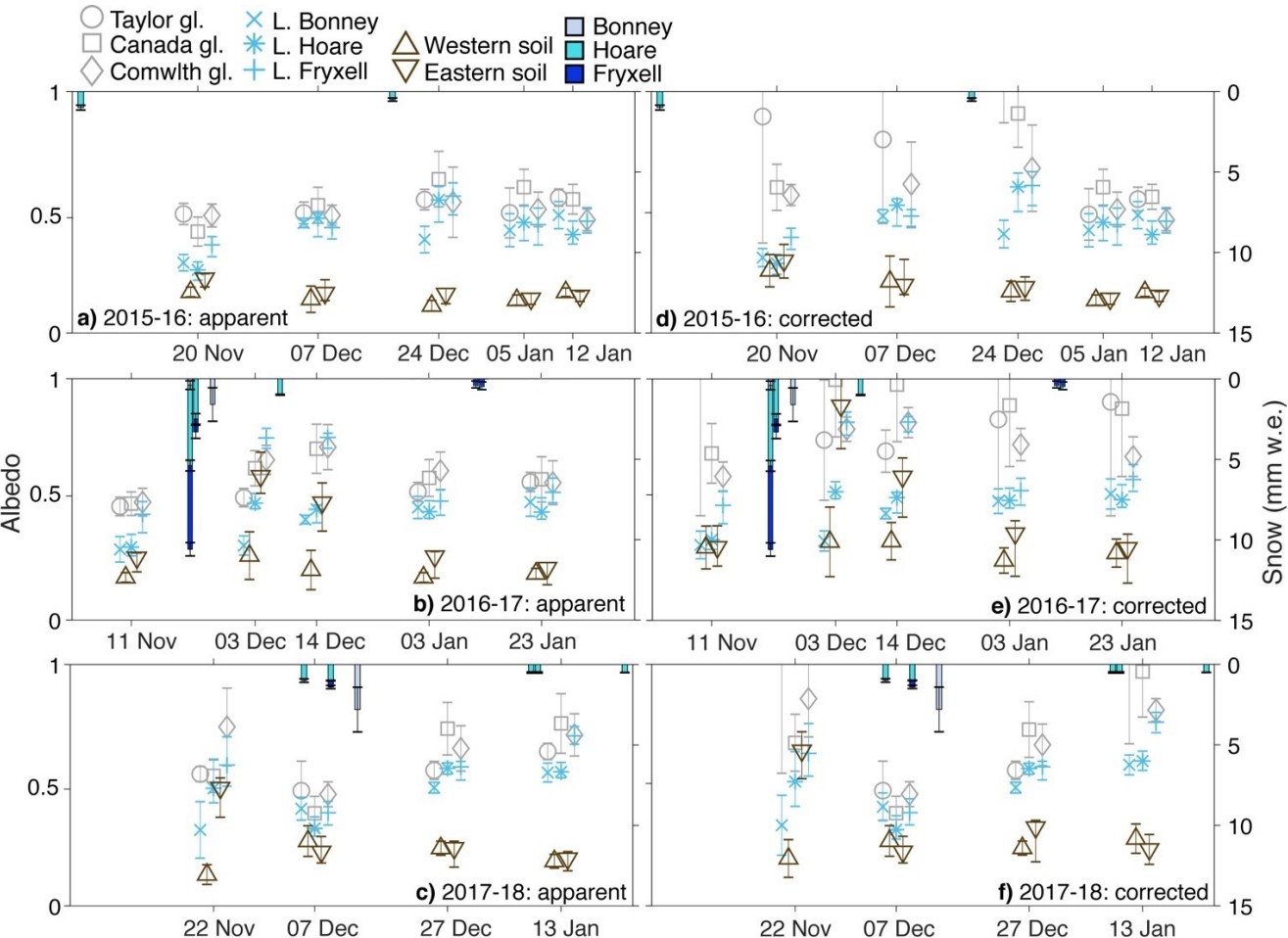

Figure 5. Mean and one standard deviation of albedo measured over the three main glaciers and lakes, and soil in the western (Bonney and Hoare basins) and eastern (Fryxell basin) Taylor Valley during each of the flights. Apparent albedo is shown in a) 2015-16 b) 2016-17 and c) 2017-18 seasons. Corrected albedo is shown in d) 2015-16 e) 2016-17 and f) 2017-18 seasons.

5    Hanging bars in all plots are snowfall measured at the Bonney, Hoare, and Fryxell met stations in mm of water equivalent.

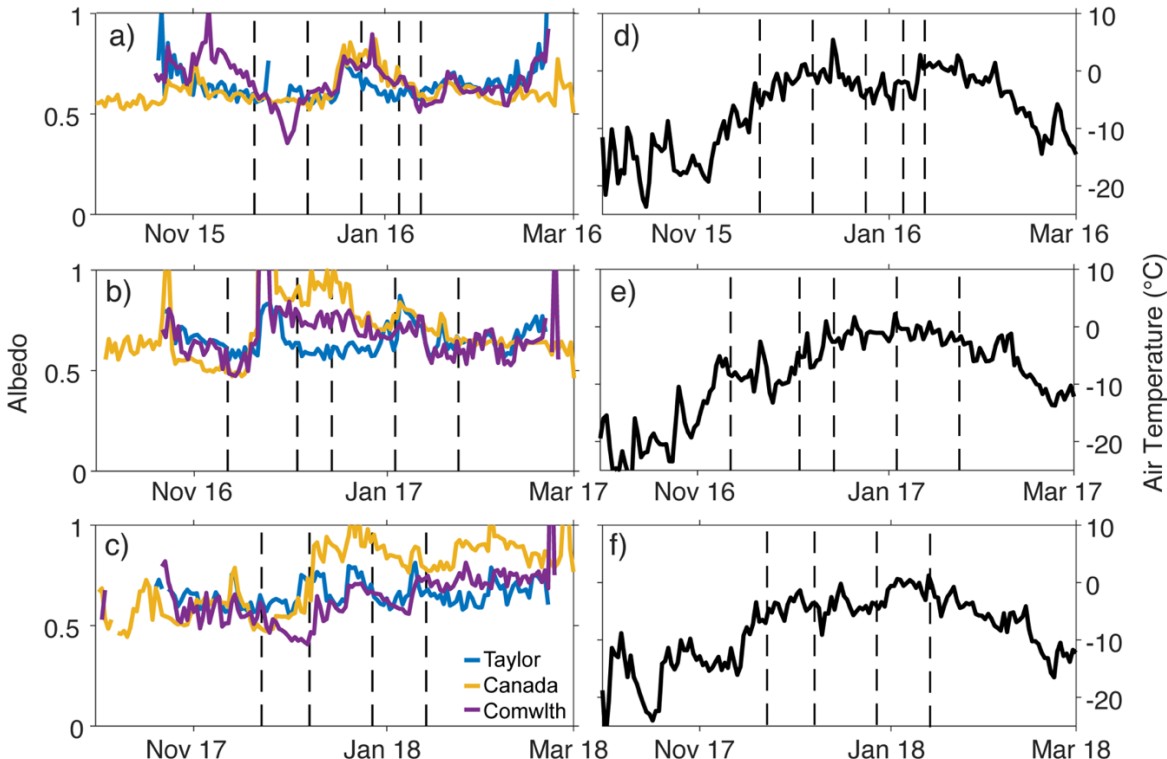

Figure 6. Timeseries of albedo measured at Taylor, Canada, and Commonwealth meteorological stations over the a) 2015-16, b) 2016-17 and c) 2017-18 seasons. Time series of air temperature measured at the Lake Hoare station of the d) 2015-16, e) 2016-17 and f) 2017-18 seasons. Lake Hoare has the best temperature record and is representative of temperature across the entire valley. Vertical dashed black lines on all figures are the times are which airborne albedo measurements were made.

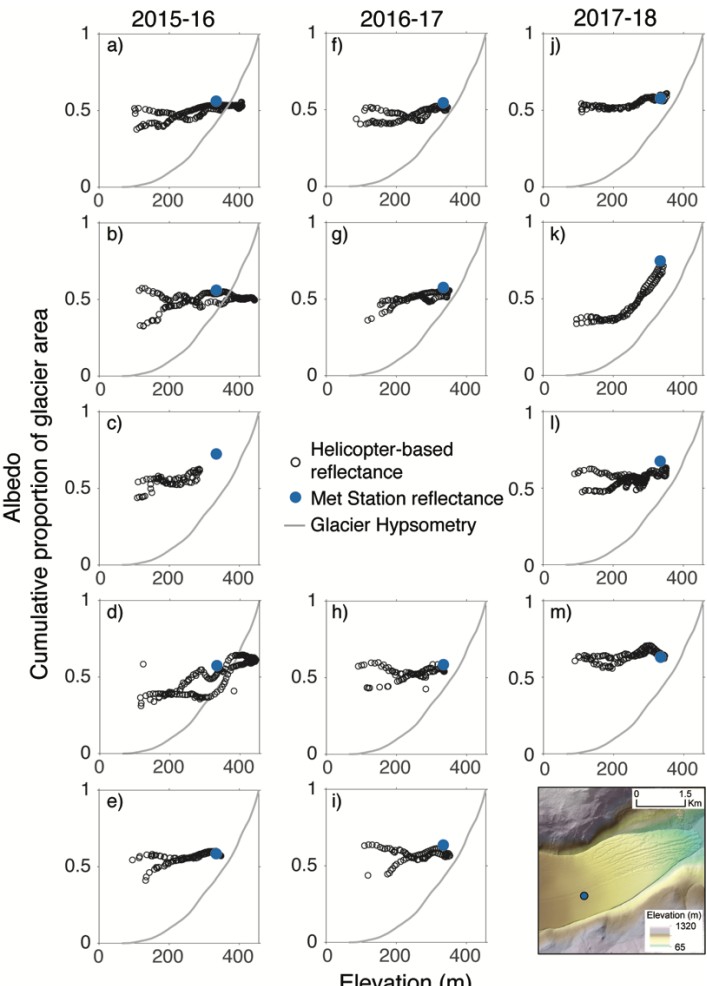

Figure 7. Apparent albedo plotted against surface elevation of Taylor Glacier for all flights in the 2015-16 season (a-e) 2016-17 season (f-i) and 2017-18 season (j-m). Black open circles are helicopter-based apparent albedo. Blue filled circles are the meteorological station albedo measured at the time of the flight. Grey line is glacier hypsometry. Map in bottom corner is a hillshade of a 1m lidar- derived digital elevation model collected in Dec 2015 with colormap of elevation. The third flight of the 2016-17 season did not cover Taylor Glacier and therefore no figure for that flight was made.

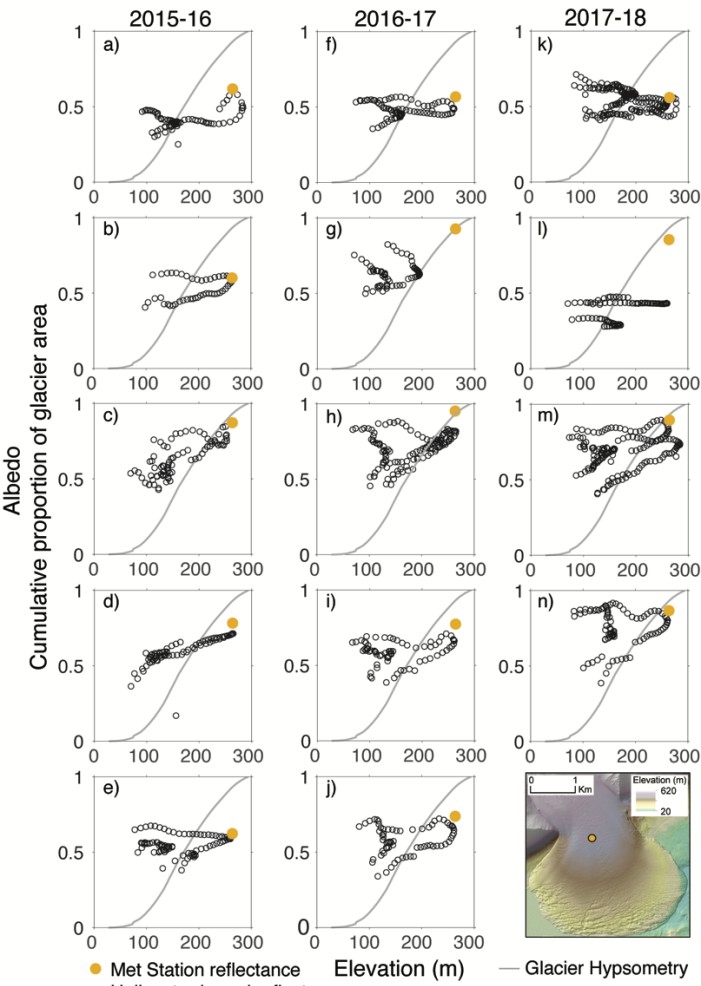

Figure 8. Apparent albedo plotted against surface elevation of Canada Glacier for all flights in the 2015-16 season (a-e) 2016-17 season (f-j) and 2017-18 season (k-n). Black open circles are helicopter-based apparent albedo. Yellow filled circles are the meteorological station albedo measured at the time of the flight. Grey line is glacier hypsometry. Grey line is glacier hypsometry. Map in bottom corner is a hillshade of a 1m lidar- derived digital elevation model collected in Dec 2015 with colormap of elevation.

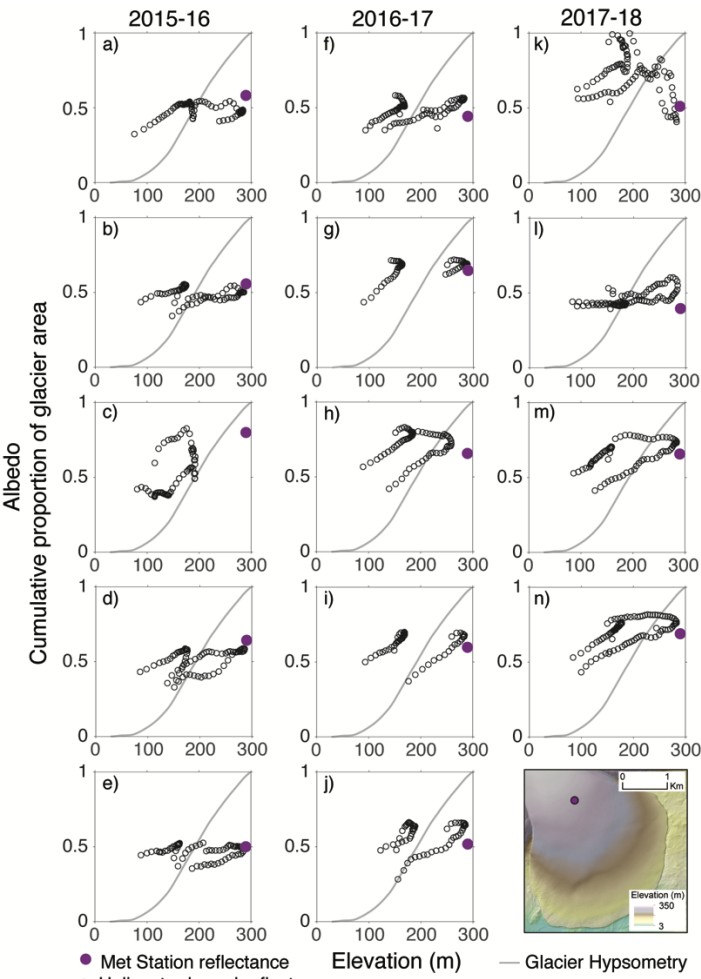

Figure 9. Apparent albedo plotted against surface elevation of Commonwealth Glacier for all flights in the 2015-16 season (a-e) 2016-17 season (f-j) and 2017-18 season (k-n). Black open circles are helicopter-based apparent albedo. Purple filled circles are the meteorological station albedo measured at the time of the flight. Grey line is glacier hypsometry. Map in bottom corner is a hillshade of a 1m lidar- derived digital elevation model collected in Dec 2015 with colormap of elevation.

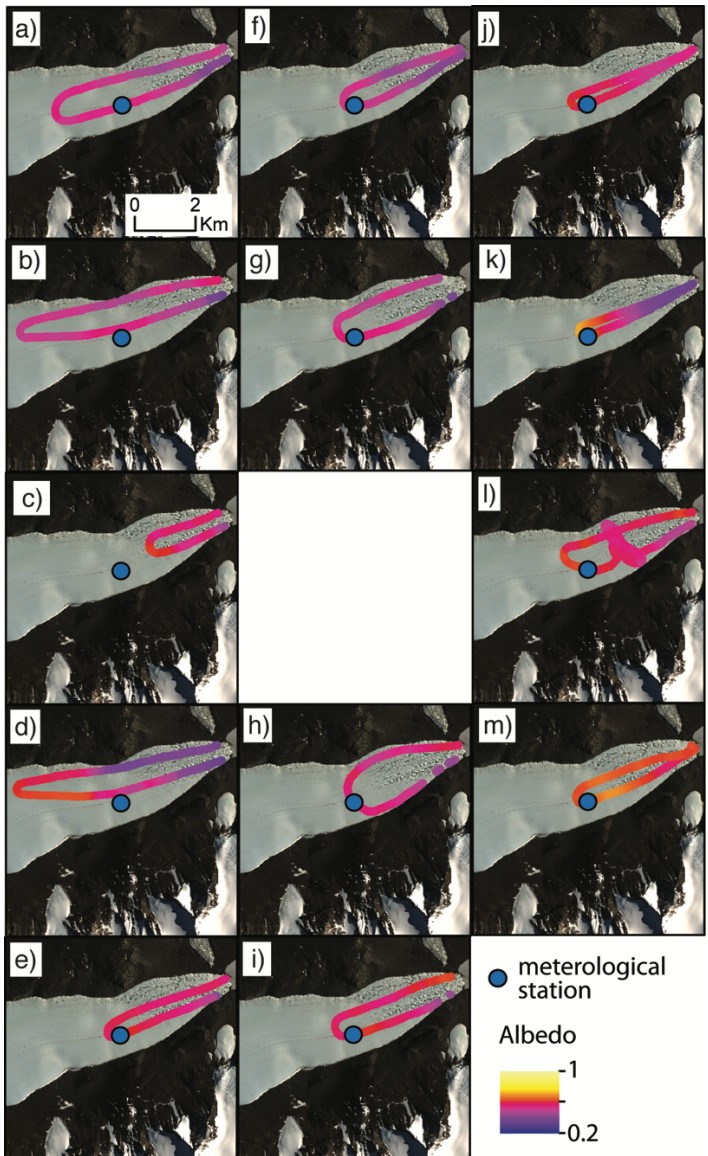

Figure 10. Flight paths of each flight on Taylor Glacier in the 2015-16 season (a-e) 2016-17 season (f-i) and 2017-18 season (j-m). Points are locations of measurements made along the flight path color scaled by apparent albedo measured at that location. Background imagery is provided by USGS/NASA; Landsat8 image collected on 14 January, 2016.

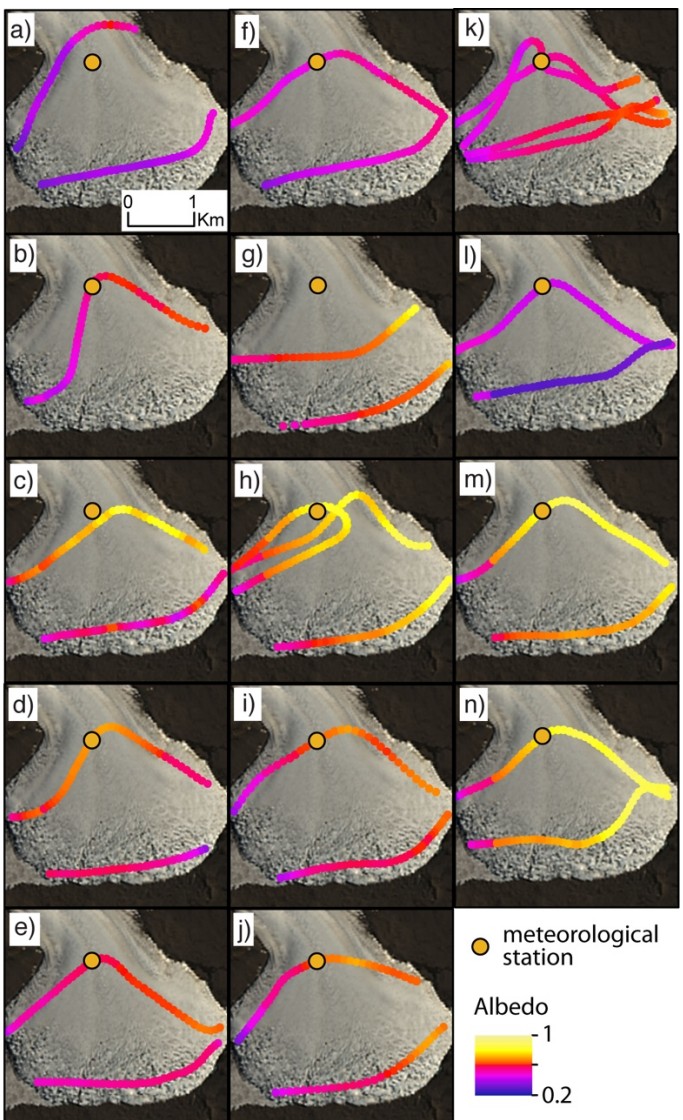

Figure 11. Flight paths of each flight on Canada Glacier in the 2015-16 season (a-e) 2016-17 season (f-j) and 2017-18 season (k-n). Points are locations of measurements made along the flight path color scaled by apparent albedo at that location. Background imagery is provided by USGS/NASA; Landsat8 image collected on 14 January, 2016.

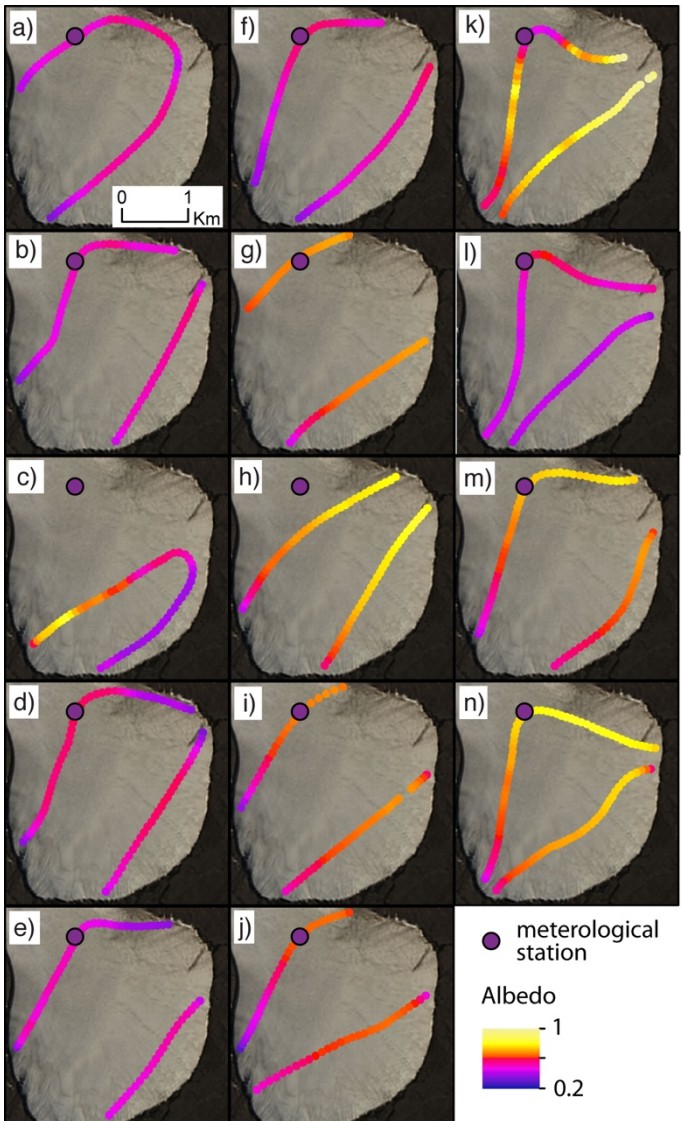

Figure 12. Flight paths of each flight on Commonwealth Glacier in the 2015-16 season (a-e) 2016-17 season (f-j) and 2017-18 season (k-n). Points are locations of measurements made along the flight path color scaled by apparent albedo measured at that location. Background imagery is provided by USGS/NASA; Landsat8 image collected on 14 January, 2016.

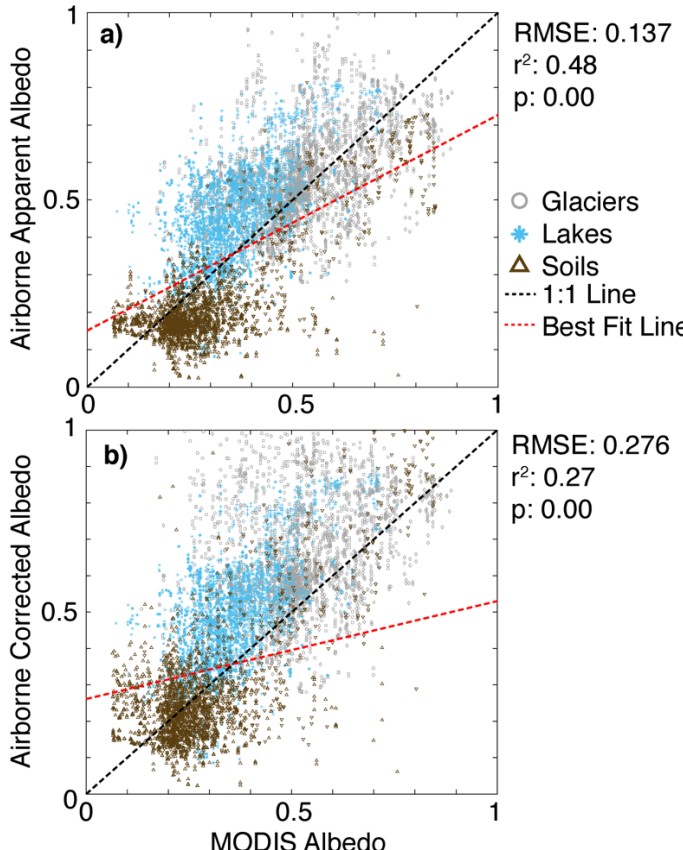

Figure 13. Comparison of all airborne albedo measurements to MODIS data across all surface types. We compare both a) apparent albedo and b) corrected albedo to MODIS data. Black dashed line is a 1:1 line, and red dashed line is the best fit line of the data.