# Peer review of "The seasonal evolution of albedo across glaciers and the surrounding landscape of the Taylor Valley, Antarctica"

_The Cryosphere, 2019_

## Referee Comment (RC1) · Anonymous Referee #1 · 17 Oct 2019

General

The authors present a study of airborne surface reflectance measurements over different landscape classes in the McMurdo Dry Valley, yielding at characterizing the albedo variability over the ablation zone of glaciers during summer. The manuscript lacks a clear hypothesis and presents a useful dataset for the region instead. The measurements were carried out thoroughly and the relation of the results to in situ observations at weather stations is meaningful. However, I have two strong suggestions to the authors, which should be implemented in the paper.

First, for the authors aim to characterize the temporal and spatial variability of albedo,

the exclusion of satellite remote sensing data (P2, L32-33 and P3, L3-5) is not justified enough. Therefore, I strongly recommend comparing the measurements to an independent dataset of spatial albedo, which is e.g. the albedo product from MODIS. There might be lags between the acquisition dates, issues with spatial resolution (maybe even a just a few available pixels), cloud cover and high solar zenith angles. However, the MODIS albedo product is widely used in Polar science and discussing space vs airborne albedo measurements would bring additional benefit to the manuscript and to the robustness of the dataset.

Second, in the discussion the authors give various reasons for the peculiarities of their results. Unfortunately, most of the reasons are not well quantified, due to lacking field evidence and thus, remain a bit speculative. I clearly miss a discussion how the high proportion of sublimation of up to 80% of total glacier ablation (Lewis et al., 1998) could impact spatial or temporal albedo variations (Lhermitte et al., 2014; Winkler et al., 2009). Since the authors have in situ measurements from weather stations, they could estimate conditions for sublimation from the difference of the vapour pressure or the dew point temperature between the surface and the near surface air layer, at times when the surface temperature is below $0°$. Consequently, it would be possible to distinguish between melt and sublimation events in the days or weeks prior to the albedo acquisition.

Structure

Although the authors state to focus on glacier surfaces, their analyses often concern lake and soil surfaces. I think this is valid as the airborne measurements potentially increase our knowledge of albedo variations within the McMurdo Dry Valley compared to other remote sensing data due to a higher spatial resolution. However, in the discussion the three different surface types are often mixed and it is not always clear which one is addressed. Maybe clearer structure in the discussion or a synoptic table in the conclusion could clarify this point.

Interactive
comment

Discussion

Chapter 4.1 "seasonal shift in albedo without the presence of snow" possibly applies an incorrect analogy. The authors focus their analyses on glacier surfaces, but use ice aging and structural changes of lake ice as an explanation for their findings. Lake ice and glacier ice have a different genesis, different deformation and recrystallization fabrics. At least for glacier surfaces I doubt that this analogy holds, especially as the relationship is just qualitatively. A more rigorous assessment applying this method on glacier ice would strengthen the manuscript.

For the discussion of spatial and temporal albedo patterns in a wider Antarctic context, I miss a connection to the results of Pirazzini (2004).

Figures

Figures 1 and 2 are too small to read accurately. Especially in Figure 2 b+c the reader can not distinguish the colour of the points and the buffers. In Fig 2c it is not clear that the Thiessen polygons really connect the measurements to the closest meteorological station. It seems that some points in the purple part would be closer to the green part.

Specific comments

P4, L12-15: How was a level measurement of the surface reflectance assured? Was the weighted box equipped with tilt meters and was a correction of the data applied (Weiser et al., 2016)? P4, L25-27: Could you add the solar zenith angle of your acquisition time? P5, L28: Could you add the slope of the location of the weather stations? P6, L7 and caption of Figure 2: Correct spelling is Thiessen. P6, L9 and throughout the manuscript: I assume you can delete percent as either the unit of the given number states it, or like in Figure 4 or 5 dimensionless values are shown. P7, L16 and Figure 3: There is also a high variation in incoming shortwave radiation on 7 Dec 2015, 22 Nov 2017 and 7 Dec 2017. What happened on these dates? Section 3.2: Could you add an explanation, if the lakes were permanently frozen, or if there were open water areas
or melt ponds? P12, L8: You should be able to estimate melt and refreezing conditions from the weather stations.

References

Lewis, K. J., Fountain, A. G. and Dana, G. L.: Surface energy balance and meltwater production for a Dry Valley glacier, Taylor Valley, Antarctica, Ann. Glaciol., 27, 603–609, 1998. Lhermitte, S., Abermann, J. and Kinnard, C.: Albedo over rough snow and ice surfaces, Cryosph., 8(3), 1069–1086, doi:10.5194/tc-8-1069-2014, 2014. Pirazzini, R.: Surface albedo measurements over Antarctic sites in summer, J. Geophys. Res., 109(D20), D20118, doi:10.1029/2004JD004617, 2004. Weiser, U., Olefs, M., Schöner, W., Weyss, G. and Hynek, B.: Correction of broadband snow albedo measurements affected by unknown slope and sensor tilts, Cryosph., 10(2), 775–790, doi:10.5194/tc-10-775-2016, 2016. Winkler, M., Juen, I., Mölg, T., Wagnon, P., Gómez, J. and Kaser, G.: Measured and modelled sublimation on the tropical Glaciar Artesonraju, Perú, Cryosph., 3(1), 21–30, doi:10.5194/tc-3-21-2009, 2009.
* * *

---

## Referee Comment (RC2) · Stephen Warren (Referee) · 6 Nov 2019

Stephen Warren (Referee)

sgw@uw.edu

Major comment:

Further analysis is required to take into account the slopes of the glacier surfaces. Although the authors did not state this explicitly, I think the radiometers at the ground stations were probably leveled horizontal, rather than parallel to the surface. If the ice surface is sloping to the south, then at midday it is receiving less incident solar flux than a horizontally-leveled upward-looking radiometer, and the measured albedo needs to be corrected for this slope. An example of the bias that can result, if the slope-correction is not made, was shown in Figure 9 of Grenfell et al. (1994) and

related discussion. A similar correction must be made to the upward flux measured from the helicopter, using the sun azimuth and elevation for the times that the different glaciers were overflown. For flights under overcast cloud, no correction is needed, because both the upward and downward radiation are diffuse.

Some considerations for how to make albedo measurements from helicopter were discussed by Allison et al. (1993), which the authors might like to read.

Minor comments:

Some confusion results from the terminology. Going "up-valley" sounds like going to higher elevation, so I was at first puzzled to read that albedo increased with increasing elevation but also increased going down-valley. Maybe you could instead say "down-Taylor" and "up-Taylor" to forestall this confusion.

p 4 line 17. Explain why the radiometer was hanging so far (6 meters) below the helicopter. How was it maintained level?

p 6 line 4-5. "Accumulation due to foehn events is removed." Why?

p 8 line 13. If the soil became damp or wet, this would explain the reduced albedo, as shown by Bøggild et al (2010, Figure 6) and explained by Bohren (1987).

p 14 line 27. "arguably". Is this word needed? Who argues against this claim?

Figure 1. In the inset, the Ross Sea is to the left of the star, but it's to the right of the main map, causing confusion. The inset should be rotated 180 degrees, so that north (at the star) is toward the top.

Figure 2 is too small. I don't see the "polygons" (line 7); maybe they will be apparent when you expand the figure.

Figure 3g. The peak seems to be at 14:00. What time zone are you using? It would be better to use local sun-time for this plot.

In Figures 4 and 5, the data are classified first by year and second by surface type. Consider reversing this hierarchy, or maybe add two figures with the reversed hierarchy. The years are different from each other, but the surface types are more different from each other than the years are. So try merging all three years onto one graph to plot the seasonal cycle for soil, for example. This will also resolve the seasonal cycle better, with 14 points from November to January instead of only 4 or 5.

Figure 4 caption line 1. Change "percent" to "fractional". Also on Figure 5.

Figure 5. Most of the information in this figure (except for the snowfall events) seems to duplicate Figure 4. Readers looking back and forth between Figure 4 and Figure 5 will be frustrated trying to make sense of any differences.

Syntax and spelling:

It is jarring to read "We" six times in the abstract. Some of these can be replaced. For example, you could say "The seasonal evolution is yet to be fully characterized", "A camera, gps, and shortwave radiometer were hung from a helicopter . . . ", "These data are coupled with incoming radiation . . . " Your sentence "We also observed that albedo followed a pattern . . . " can be shortened to "The albedo followed a pattern . . . "

page 3 line 31. "wind-transported material that frequently melts to form cryoconite holes". It is the ice that melts, not the wind-transported material.

page 5 line 3. " . . . did not meet usability standards or associated with . . . "

p 9 line 16. "on the that flight"

p 9 line 31. Change "principals" to "principles".

p 16 line 22. "This research a part of"

p 17 line 4. Change Antarcitca to Antarctica.

p 19 line 13. Change Stoeve to Stroeve.

p 19 line 28. Change "Manag." to "Research"

p 24 line 4-5 "shown are as". Maybe you mean "are shown as".

References:

Allison, I., R.E. Brandt, and S.G. Warren, 1993: East Antarctic sea ice: albedo, thickness distribution and snow cover. J. Geophys. Res. (Oceans), 98, 12417-12429.

Bøggild, C.E., R.E. Brandt, K.J. Brown, and S.G. Warren, 2010: The ablation zone in northeast Greenland: Ice types, albedos, and impurities. J. Glaciol., 56, 101-113.

Bohren, C.F., 1987: Multiple scattering at the beach. Chapter 12 of Clouds in a Glass of Beer, Simple Experiments in Atmospheric Physics. Wiley, New York, 195 pp.

Grenfell, T.C., S.G. Warren, and P.C. Mullen, 1994: Reflection of solar radiation by the Antarctic snow surface at ultraviolet, visible, and near-infrared wavelengths. J. Geophys. Res., 99, 18669-18684.

---

## Author Comment (AC1) · 4 Dec 2019

General
The authors present a study of airborne surface reflectance measurements over different landscape classes in the McMurdo Dry Valley, yielding at characterizing the albedo variability over the ablation zone of glaciers during summer. The manuscript lacks a clear hypothesis and presents a useful dataset for the region instead. The measurements were carried out thoroughly and the relation of the results to in situ observations at weather stations is meaningful. However, I have two strong suggestions to the authors, which should be implemented in the paper.

First, for the authors aim to characterize the temporal and spatial variability of albedo, the exclusion of satellite remote sensing data (P2, L32-33 and P3, L3-5) is not justified enough. Therefore, I strongly recommend comparing the measurements to an independent dataset of spatial albedo, which is e.g. the albedo product from MODIS. There might be lags between the acquisition dates, issues with spatial resolution (maybe even a just a few available pixels), cloud cover and high solar zenith angles. However, the MODIS albedo product is widely used in Polar science and discussing space vs airborne albedo measurements would bring additional benefit to the manuscript and to the robustness of the dataset.
We have taken this suggestion and have incorporated the MODIS albedo product data in our study in order to compare our albedo measurements with MODIS. We have added a section into the methods describing how we acquired these data, a figure in the manuscript comparing all measurements to MODIS, two figures in the supplemental information comparing both apparent and corrected albedo to MODIS for each flight, and a section in the discussion about our findings from the comparison.

Second, in the discussion the authors give various reasons for the peculiarities of their results. Unfortunately, most of the reasons are not well quantified, due to lacking field evidence and thus, remain a bit speculative. I clearly miss a discussion how the high proportion of sublimation of up to 80% of total glacier ablation (Lewis et al., 1998) could impact spatial or temporal albedo variations (Lhermitte et al., 2014; Winkler et al., 2009). Since the authors have in situ measurements from weather stations, they could estimate conditions for sublimation from the difference of the vapour pressure or the dew point temperature between the surface and the near surface air layer, at times when the surface temperature is below 0◦ . Consequently, it would be possible to distinguish between melt and sublimation events in the days or weeks prior to the albedo acquisition.
We have added a figure (now figure 6) including a timeseries of air temperature in order to better discuss the energy balance and how this relates to changes in albedo. We believe that calculating

the energy balance conditions that lead to sublimation and melt require more complex modeling that is beyond the scope of this study (i.e. Lewis et al., 1998, Hoffman et al., 2008, 2014, 2016). Modeling work for this time period is underway and we plan on publishing results of this work that include the relationships and feedbacks between albedo and ablation.
We do agree that sublimation likely does affect albedo and have added sentences to the discussion addressing that (now p 14 lines 20-27).

Structure
Although the authors state to focus on glacier surfaces, their analyses often concern lake and soil surfaces. I think this is valid as the airborne measurements potentially increase our knowledge of albedo variations within the McMurdo Dry Valley compared to other remote sensing data due to a higher spatial resolution. However, in the discussion the three different surface types are often mixed and it is not always clear which one is addressed. Maybe clearer structure in the discussion or a synoptic table in the conclusion could clarify this point.
We agree that certain parts of the manuscript mix surface type and this discussion may be confusing. We have edited throughout to try and clarify this. We believe the biggest source of confusion was in section 3.2 (now 4.2). This section has been edited heavily with most of that language removed. We thank the reviewer for this suggestion and believe the manuscript is now more clear.

Discussion
Chapter 4.1 "seasonal shift in albedo without the presence of snow" possibly applies an incorrect analogy. The authors focus their analyses on glacier surfaces, but use ice aging and structural changes of lake ice as an explanation for their findings. Lake ice and glacier ice have a different genesis, different deformation and recrystallization fabrics. At least for glacier surfaces I doubt that this analogy holds, especially as the relationship is just qualitatively. A more rigorous assessment applying this method on glacier ice would strengthen the manuscript.
We have been very explicit in the manuscript that while both lakes and glaciers have ice whitening, the ice structure is different between them and the exact ways in which ice albedo changes are a function of that difference. We discuss the previous research on lake ice in the second and third paragraphs of section 4.1 (now 5.1). We discuss what we know about ice whitening in glacier ice from previous research in the fourth, fifth, and sixth paragraphs of that section. We also state twice in this section that there are differences between lake and glacier ice. We respectfully disagree with this comment and believe that we adequately separate seasonal changes in albedo between lakes and glaciers and do not attempt to apply research on lakes to glaciers.

For the discussion of spatial and temporal albedo patterns in a wider Antarctic context, I miss a connection to the results of Pirazzini (2004).
We have included comparisons to this study throughout the manuscript. We appreciate the reviewer bringing this paper to our attention. It has added valuable discussion material to this manuscript.

Figures
Figures 1 and 2 are too small to read accurately. Especially in Figure 2 b+c the reader can not distinguish the colour of the points and the buffers. In Fig 2c it is not clear that the Thiessen

polygons really connect the measurements to the closest meteorological station. It seems that some points in the purple part would be closer to the green part.

Figures 1 and 2 have been enlarged. We believe that all details of each figure are now discernable.

Specific comments

P4, L12-15: How was a level measurement of the surface reflectance assured? Was the weighted box equipped with tilt meters and was a correction of the data applied (Weiser et al., 2016)?

We observed the box throughout the flight on most flights and believe the box was maintained level in flight due to the weight and the fin incorporated in the box design. Unfortunately tilt meters were not available to us and we do not have those data to be able to make those corrections. We do add language acknowledging this, discuss how swing can be a potential source of error, and estimates of the error adapted from Allison et al. (1993) as per the suggestion of reviewer 2. This can be found in the new section we added titled "Error sources and albedo correction"

P4, L25-27: Could you add the solar zenith angle of your acquisition time?

We added a sentence at that location stating the range of solar zenith angles (54.3 – 61.8 degrees) across all flights.

P5, L28: Could you add the slope of the location of the weather stations?

The meteorological stations are all leveled and are measuring reflected radiation on a flat part of the glacier (slope = 0°). We have added this to the text to clarify this fact.

P6, L7 and caption of Figure 2: Correct spelling is Thiessen.

This has been corrected in both locations.

P6, L9 and throughout the manuscript: I assume you can delete percent as either the unit of the given number states it, or like in Figure 4 or 5 dimensionless values are shown.

We have changed all text and figures so albedo ranges from 0 to 1 and use of percent has been removed from the manuscript. We thank the reviewer for calling this inconsistency to our attention.

P7, L16 and Figure 3: There is also a high variation in incoming shortwave radiation on 7 Dec 2015, 22 Nov 2017 and 7 Dec 2017. What happened on these dates?

The variation is due to clouds on those days. We attempted to makes all flights on uniformly sunny or cloudy days while balancing helicopter availability, weather, and well-spaced data acquisition throughout the season. We did not have perfect conditions on those days due to a stretch of cloudy days (Figure 2) and limited helicopter availability because of weather delays. We are able to mostly control for this by using the closest meteorological station for incoming radiation data.

Section 3.2: Could you add an explanation, if the lakes were permanently frozen, or if there were open water areas or melt ponds?

The lakes are permanently frozen across the majority of their area. Moats of open water melt around the shoreline of the lakes however, they are less than 100m wide and therefore are not included in this analysis.
Canada and Taylor Glaciers have melt ponds that also have a permanent ice cover.
We add some clarifying language to this section to indicate that no measurements are made over open water for either glaciers or lake ice.

P12, L8: You should be able to estimate melt and refreezing conditions from the weather stations.
This section is referring to the process of refreeze of meltwater as it is moving through the ice matrix and supraglacial stream network. If meltwater is generated high on the glacier and far from a supraglacial channel (i.e. long travel time) it is highly likely that it will refreeze before it leaves the glacier. Melt is generated in the subsurface due to internal heating from penetrating shortwave radiation, rather than sensible heat. This melt and subsequent refreeze process is rather intensive to calculate and cannot easily be estimated from meteorological stations. For this reason, we believe it is beyond the scope of this paper. This process has been modeled however and we believe it is sufficient to cite that work to support our hypotheses here. A citation has been added to this line to clarify that this process has been identified for these glaciers.

---

## Author Comment (AC2) · 4 Dec 2019

Stephen Warren (Referee)
sgw@uw.edu

Major comment: Further analysis is required to take into account the slopes of the glacier surfaces. Although the authors did not state this explicitly, I think the radiometers at the ground stations were probably leveled horizontal, rather than parallel to the surface. If the ice surface is sloping to the south, then at midday it is receiving less incident solar flux than a horizontally-leveled upward-looking radiometer, and the measured albedo needs to be corrected for this slope. An example of the bias that can result, if the slope-correction is not made, was shown in Figure 9 of Grenfell et al. (1994) and related discussion. A similar correction must be made to the upward flux measured from the helicopter, using the sun azimuth and elevation for the times that the different glaciers were overflown. For flights under overcast cloud, no correction is needed, because both the upward and downward radiation are diffuse.

We appreciate the reviewer making this point. We have done extra analysis taking this concept in to account coupled with an additional literature review. The meteorological stations are all leveled and measuring albedo over a horizontal surface and therefore do not need correction. We applied the method outlined by Grenfell et al. (1994) to our airborne dataset of apparent albedo. This method works well for most locations over which we collected data. However certain areas, namely the lower elevations of Canada and Taylor glacier, are too topographically complex for this method to work well and produces physically unrealistic results. We tested corrections using both and mean and median slope with little difference between the results. We did an additional literature review to determine how this problem is addressed elsewhere and found that methods vary depending on if they were developed for station or satellite data and unfortunately nether scale is particularly suitable for this application. We discuss limitations of the simplified correction method and locations where it is particularly problematic in a new section we added to the manuscript titled "Error sources and albedo correction"

Some considerations for how to make albedo measurements from helicopter were discussed by Allison et al. (1993), which the authors might like to read.
Discussion of this paper is included in the error sources and estimation section that was added to the manuscript. We thank the reviewer for making us aware of this paper. It was very valuable in developing this section.

Minor comments: Some confusion results from the terminology. Going "up-valley" sounds like going to higher elevation, so I was at first puzzled to read that albedo increased with increasing elevation but also increased going down-valley. Maybe you could instead say "downTaylor" and "up-Taylor" to forestall this confusion.

We changed this language throughout and now use west, generally referring to the Hoare and Bonney basins closer to the polar plateau, and east, generally referring to the Fryxell basin and closer to the Ross Sea.

p 4 line 17. Explain why the radiometer was hanging so far (6 meters) below the helicopter. How was it maintained level?

The radiometer had to be in a box slung from the helicopter due to safety regulations of the helicopter contractor. We explored the possibility of mounting it directly to the helicopter, but this was not possible due to rules in place about what can be attached to the helicopter. The solution was to sling the instruments. They are maintained level by over 200 lbs of ballast in the box and a large fin attached to the back of box. There is minor swing of the box however, observations of the flights lead us to believe that it was maintained mostly level throughout the flight. We add language discussing how swing may contribute to error in new section titled "Error sources and albedo correction"

p 6 line 4-5. "Accumulation due to foehn events is removed." Why?

This statement was made in error. Accumulation due to foehn events was removed in processing this dataset for other purposes but was not done here. Any accumulation of snow will affect albedo and wind-driven accumulation is left in the dataset presented in this study. This sentence has been removed

p 8 line 13. If the soil became damp or wet, this would explain the reduced albedo, as shown by Bøggild et al (2010, Figure 6) and explained by Bohren (1987).

We appreciate the reviewer bringing this to our attention and agree that soil wetting likely reduces albedo due to regular deliquescence of MDV soils. We have added language discussing this as a final paragraph in section 5.1

p 14 line 27. "arguably". Is this word needed? Who argues against this claim?

Arguably has been removed and the sentence has been edited. It now reads:

Albedo is one of the most important parameters for glacial energy balance across all types of glaciers at all latitudes

Figure 1. In the inset, the Ross Sea is to the left of the star, but it's to the right of the main map, causing confusion. The inset should be rotated 180 degrees, so that north (at the star) is toward the top.

We have rotated the map and it is now in the same orientation as the main map.

Figure 2 is too small. I don't see the "polygons" (line 7); maybe they will be apparent when you expand the figure.

Figure 2 has been enlarged. The polygons should now be visible just inside the edge of the lakes and glaciers.

Figure 3g. The peak seems to be at 14:00. What time zone are you using? It would be better to use local sun-time for this plot.

We have adjusted this and the plot now uses local sun time.

In Figures 4 and 5, the data are classified first by year and second by surface type. Consider reversing this hierarchy, or maybe add two figures with the reversed hierarchy. The years are different from each other, but the surface types are more different from each other than the years are. So try merging all three years onto one graph to plot the seasonal cycle for soil, for example. This will also resolve the seasonal cycle better, with 14 points from November to January instead of only 4 or 5.

See response to comment below.

Figure 4 caption line 1. Change "percent" to "fractional". Also on Figure 5.

This has been changed in both figures.

Figure 5. Most of the information in this figure (except for the snowfall events) seems to duplicate Figure 4. Readers looking back and forth between Figure 4 and Figure 5 will be frustrated trying to make sense of any differences.

We have changed both figures 4 and 5. Figure 4 now focuses on the range of corrected values across landscape types. All seasons are on the same figure and figures are separated by glaciers, lakes, and soils. Figure 5 is similar to what was in the original submitted manuscript, but a new panel has been added that is the same three figures separating individual, glaciers, lakes, and soils, across seasons but uses corrected albedo instead of apparent albedo. This new figure configuration allows us to discuss differences across each of these landscape components using both correct and apparent albedo. We believe that this improves discussion and illustrates the differences between apparent and corrected albedo (Figure 5) and how this changes across landscape types, while showing the overall seasonal patterns of albedo (Figure 4).

Syntax and spelling:
It is jarring to read "We" six times in the abstract. Some of these can be replaced. For example, you could say "The seasonal evolution is yet to be fully characterized", "A camera, gps, and shortwave radiometer were hung from a helicopter . . . ", "These data are coupled with incoming radiation . . . " Your sentence "We also observed that albedo followed a pattern . . . " can be shortened to "The albedo followed a pattern . . . "

These edits have been made as well as some additional edits to the abstract to improve clarity. We believe the abstract is now more clear and readable.

page 3 line 31. "wind-transported material that frequently melts to form cryoconite holes". It is the ice that melts, not the wind-transported material.

This has been edited to clarify that the material melts the ice below it.

page 5 line 3. " . . . did not meet usability standards or associated with . . . "

We added "were" between or and associated. The sentence now reads: Individual measurements were either discarded if they did not meet usability standards or were associated with a given landscape feature and the closest meteorological station

p 9 line 16. "on the that flight"

We removed the word "the"

p 9 line 31. Change "principals" to "principles".

This change has been made

p 16 line 22. "This research a part of"
We added "is" to the sentence. It now reads: This research is a part of the McMurdo LTER, funded by the National Science Foundation grants 1115245 and 1637708.

p 17 line 4. Change Antarcitca to Antarctica.
This has been changed.

p 19 line 13. Change Stoeve to Stroeve.
This has been changed.

p 19 line 28. Change "Manag." to "Research"
This has been changed.

p 24 line 4-5 "shown are as". Maybe you mean "are shown as".
This sentence has been fixed to read "are shown as".